# WIDE NEURAL NETWORK TRAINING DYNAMICS FOR REINFORCEMENT LEARNING

## ABSTRACT

While deep reinforcement learning (RL) has demonstrated remarkable empirical success, understanding certain aspects of the training of deep RL agents remains elusive, leading to many RL algorithms requiring additional heuristic ingredients to be practically useful. In contrast to supervised learning, RL algorithms typically do not have access to ground-truth labels, leading to a more challenging training setup. In this work, we analyze the training dynamics of overparametrized, infinitely-wide value function networks, trained through temporal difference updates by extending previous results from neural tangent kernel approaches in supervised learning. We derive closed-form expressions for the training dynamics of common temporal difference policy evaluation methods as well as an analysis on the effects of uncertainty quantification of ensembling, a common heuristic measure of uncertainty in RL, in the infinite-width limit. We validate our analytically derived dynamic predictions on a toy environment, where we find good agreement with real neural networks. We also evaluate our methods on the classic control cart pole environment, and discover that the predictions and uncertainty quantification of our analytical solutions outperform those made by true ensembles trained via gradient descent.

## 1 INTRODUCTION

Deep reinforcement learning (RL) presents great promise as a tool for sequential decision-making in complex environments, although it comes with a number of unique challenges compared to other machine learning tasks. For example, common features of deep RL algorithms are function approximation, bootstrapping, and off-policy learning – sometimes referred to as *the deadly triad* (Sutton & Barto, 2018) – which can lead to instability in training. This manifests in practice in the form of algorithms that require significant fine-tuning of numerous hyperparameters and additional ingredients required to make training stable.

The theoretical tractability of the training mechanics of deep RL is also arguably more challenging than that of deep supervised learning. Well-established branches of theory exist in supervised learning that model the training dynamics of neural networks (NNs) trained with supervised data, such as the neural tangent kernel (NTK) approach (Jacot et al., 2018; Huang & Yau, 2020; Zhang et al., 2017; Allen-Zhu et al., 2019). This approach has led to insight into the generalization capabilities of training neural networks (Jacot et al., 2018; Roberts et al., 2022) as well as the development of principled methods of commonly used heuristic approaches, such as ensembling for uncertainty quantification (Lee et al., 2019).

In contrast, the usual training setups in RL do not admit ground-truth labeled data and are therefore not immediately tractable with the tools developed for supervised learning. For example, a common way to train value networks in RL is to employ bootstrapping: the regression is a *moving* target, so that by the end of training, the network converges to the correct value function. We aim to bring the tools and formalism developed in the NTK approach to RL. In particular, we extend the work presented by Lee et al. (2019) to address networks trained through temporal difference (TD) updates in the infinite width limit and the effect that ensembling, a ubiquitous method used in RL for uncertainty quantification, has on the uncertainty predictions.

**Contributions.** We derive the infinite-width limit neural network training dynamics solutions for TD learning (sec. 4) considering the cases where gradients are propagated to all terms in the loss, corresponding to using a single value network (sec. 4.1), as well as the more common setup in practice that uses a primary and a target Q-network (sec. 4.2). We validate our predictions on a simple, interpretable toy environment as well as the classical control cart pole environment and compare the predictions to the empirical training of an ensemble of networks (sec. 5). We find that the analytic infinite-width limit solution of the dynamics equations captures epistemic uncertainty better than its empirical counterpart, suggesting that ensembles may behave unexpectedly or suboptimally in TD learning.

## 2 RELATED WORK

### 2.1 TEMPORAL DIFFERENCE LEARNING

Many model-free deep RL approaches rely on iteratively evaluating and improving a particular policy. Approximate dynamic programming methods rely on bootstrapping the current estimate of the value function to improve the evaluation of the learned policy's value, and this approach is crucial to many off-policy methods (Mnih et al., 2015; Lillicrap et al., 2015; Fujimoto et al., 2018; Haarnoja et al., 2018). Given a current parametric estimate $Q_\theta(s, a)$ for the expected return of policy $\pi$ after taking action $a$ in state $s$, the TD loss corresponding to a batch of transitions $(s_i, a_i, r_i, s'_i)$ corresponds to

$$\mathcal{L}(\theta) = \sum_i \left(Q_\theta(s_i, a_i) - r_i - \gamma Q_\theta(s'_i, \pi(s'_i))\right)^2, \tag{1}$$

where in practice the term involving $Q_\theta(s'_i, \pi(s'_i))$ is often replaced with a target network derived from $Q$ that does not propagate gradients to stabilize training. Bootstrapping is particularly necessary for offline RL methods, where no further interaction with the environment is possible and improving the policy via importance sampling gradients would result in excessively high variance (Precup, 2000).

Existing work in the analysis of convergence and training dynamics of value functions under TD updates with function approximators mostly limits itself to the tabular or linear function approximation settings (Bhandari et al., 2018; Gordon, 2000) due to the low degree of tractability of deep neural networks. Nonetheless, even in the linear function approximation case it can be shown that TD updates can lead to divergent training (Tsitsiklis & Van Roy, 1997), highlighting the fact that we do not yet have a good characterization of nonlinear function approximators in RL. Recent work has paid particular attention to how the learning dynamics of RL affect the neural network's learned representations (Lyle et al., 2019). We complement the investigation into the training dynamics of deep value function approximators by adapting existing techniques for the infinite-width analysis in supervised learning to TD updates for policy evaluation.

### 2.2 WIDE NEURAL NETWORKS

One of the objectives in the theory of deep learning is to explain training and generalization, which is made challenging due to the complexity of neural networks. Surprisingly, in the limit where the width of the hidden layers tends to infinity, analytical approaches become tractable (Jacot et al., 2018; Lee et al., 2019; Zhang et al., 2017; Allen-Zhu et al., 2019).

Of particular importance to our analysis is the work by Lee et al. (2019), which attempts to characterize training and generalization for infinitely wide neural networks in the supervised learning setting, and show correspondence between ensembles of wide neural networks and Gaussian processes.

The derived theoretical results are exact only in the case of neural networks with infinitely wide layers and continuous time gradient descent, although the authors find good correspondence between the infinite width case and finite-sized networks and provide critical learning rates under which discrete gradient descent steps approximate the dynamics of their continuous counterpart. In this work we have taken the main results from the work by Lee et al. (2019) and attempted to adapt these findings in the RL setting.

### 2.3 Ensembles in Reinforcement Learning

A common practical method to estimate uncertainty in value function estimates in RL is to employ an ensemble of neural networks trained in parallel through TD updates with different random initializations. Ensembling provides state-of-the-art uncertainty quantification in supervised learning (Gal & Ghahramani, 2016; Lakshminarayanan et al., 2017), and has proven to be an empirically successful approach in a wide range of RL contexts, from encouraging efficient exploration in online learning (Osband et al., 2018; 2016) to ensuring that learned policies are supported with a high degree of confidence in offline RL (An et al., 2021; Eriksson et al., 2022). As a consequence, the quality of uncertainty estimation has been identified as one of the current potential bottlenecks in improving RL algorithms (Levine et al., 2020). While successful in practice, the theoretical backing for using ensembles as a principled method for uncertainty quantification is weak. For example, it can be shown that naïvely trained ensemble members of deep neural networks need to be trained with additional terms in the loss in order to be interpreted as samples from a Bayesian posterior (D'Angelo & Fortuin, 2021), which is not widespread practice in RL. With our work, we investigate the effect of using ensembles in the infinite width limit for uncertainty quantification, and characterize the behavior of an ensemble of independently trained value functions.

## 3 Preliminaries

We consider the standard discounted reward RL setting (Sutton & Barto, 2018; Levine et al., 2020), formalized as a Markov decision process (MDP), given by a tuple $(\mathcal{S}, \mathcal{A}, r, P, \gamma)$, where $\mathcal{S}$ is the set of states, $\mathcal{A}$ the set of actions, $r = r(s, a)$ is the reward obtained after taking action $a$ from state $s$, $P(s'|s, a)$ is the probability of transitioning to state $s'$ from state $s$ with action $a$ (also called the environment transition probability, or dynamics of the system) and $\gamma \in [0, 1]$ the discount factor. We will consider the off-policy policy evaluation task: for any given policy we wish to evaluate the value of that policy from a static dataset.

The expected return following policy $\pi$ satisfies the Bellman equation for the state-action value function $Q$:

$$Q^\pi(s, a) = r + \gamma \mathbb{E}_{s' \sim P(s'|s,a),\, a' \sim \pi(a'|s')} \left[ Q^\pi(s', a') \right], \tag{2}$$

which leads to the TD update rule presented in eq. 6.

Next, we extend the notation by Lee et al. (2019) to apply to the TD setting. For the purpose of notation in this section, we will assume the action is part of the input to the Q-network (as is customary for continuous action spaces). Let $x$ be a concatenation of $s$ and $a$, so $x_i = (s_i, a_i)$ and $\mathcal{D} = \{(s_i, a_i, r_i, s'_i)\}_i$ denotes the dataset of previously collected experience that is available for training. We will use $|\mathcal{D}|$ to denote the number of transitions in the dataset, $|\theta|$ to denote the number of parameters in the neural network, $|s|$ and $|a|$ to denote the dimensions of the state and action spaces respectively. We will consider the case where $Q^\pi$ is approximated using a neural network $Q_\theta$, with parameters $\theta = \{\boldsymbol{W}^{(i)}, \boldsymbol{b}^{(i)}\}_i$ for every layer $i$, and a nonlinearity $\phi$ (such as a sigmoid or a ReLU function), for example:

$$Q_\theta(x) = \phi \left( \phi \left( \ldots \phi(x \boldsymbol{W}^{(1)} + \boldsymbol{b}^{(1)}) \ldots \right) \boldsymbol{W}^{(L-1)} + \boldsymbol{b}^{(L-1)} \right) \boldsymbol{W}^{(L)} + \boldsymbol{b}^{(L)}. \tag{3}$$

Following Lee et al. (2019) we will use the NTK parametrization:

$$\begin{cases} \boldsymbol{W}^{(l)}_{i,j} = \frac{\sigma_w}{\sqrt{n_l}} \omega^{(l)}_{i,j} \\ \boldsymbol{b}^{(l)}_j = \sigma_b \beta^{(l)}_j \end{cases},$$

where $\omega^{(l)}_{i,j}$ and $\beta^{(l)}_j$ are the trainable parameters, drawn i.i.d. from a standard Gaussian, $\omega^{(l)}_{i,j}, \beta^{(l)}_j \sim \mathcal{N}(0, 1)$, and $n_l$ is the layer width for layer $l$.

As is standard in the NTK literature, the theoretical analysis will be approached under the lens of full-batch continuous-time gradient descent. Following the work by Lee et al. (2019), from which we directly borrow notation, the base equations of motion for the vector of parameters $\theta_t$ at time $t$ with learning rate $\eta$ is

$$\dot{\theta}_t = -\eta \nabla_\theta \mathcal{L}. \tag{4}$$

The exact form of the loss $\mathcal{L}$ and how gradients are propagated through it will depend on the TD update rule modeled, and will be discussed in the next section and will depend on whether a target network is used, which is a common and often necessary ingredient to make RL algorithms stable (Mnih et al., 2015). Thus, we will differentiate between when only a single Q-network is used and where instead a target network is also employed along with the *primary network* (sometimes referred to as primary, online, behavior, or policy network).

We will find it useful to refer to $\mathcal{X}$ as a matrix of all the state-action pairs in the dataset, i.e.:

$$\mathcal{X} = \begin{bmatrix} x_1 \\ x_2 \\ \vdots \\ x_{|\mathcal{D}|} \end{bmatrix}, \quad \mathcal{X} \in \mathbb{R}^{|\mathcal{D}| \times (|s|+|a|)}$$

and $\mathcal{X}'$ for a matrix of all next-state-action pairs, where the next action $a'$ is computed from $s'$ by the policy we wish to evaluate. Similarly, $r$ is the stack of rewards $r$ from the dataset. Additionally, we will use $\bar{\theta}$ to indicate the parameters of the target network ($\theta$ for primary). For the Q-values, we will use the following notation:

$$Q^{\text{lin}}_{\theta_t}(x')\,,$$

The approximation of Q-values by linearization at state-action $x'$
using parameters $\theta_t$ at timestep $t$ (we will use $\bar{\theta}_t$ for target NN parameters).

where the linearization is a first-order Taylor expansion around $\theta = \theta_0$. Letting $\omega_t \overset{\text{def}}{=} \theta_t - \theta_0$:

$$Q^{\text{lin}}_{\theta_t}(x) \overset{\text{def}}{=} Q_{\theta_0}(x) + \nabla_\theta Q_{\theta_0}(x)|_{\theta=\theta_0}\, \omega_t\,. \tag{5}$$

We will also be employing the following compact notation:

- $Q_t \overset{\text{def}}{=} Q^{\text{lin}}_{\theta_t}(\mathcal{X}) \in \mathbb{R}^{|\mathcal{D}| \times 1}$ is the approximation of Q-values by linearization at state-actions $\mathcal{X}$ (from the dataset), using parameters $\theta$ at time $t$ ($\nabla_\theta Q_t \in \mathbb{R}^{|\mathcal{D}| \times |\theta|}$);

- $Q'_t \overset{\text{def}}{=} Q^{\text{lin}}_{\theta_t}(\mathcal{X}') \in \mathbb{R}^{|\mathcal{D}| \times 1}$ – at $\mathcal{X}'$ (each element $x' = (s', \pi(s'))$ coming from a combination of dataset and policy), using $\theta$ in the single network case; and using $\bar{\theta}$ in the primary & target network case;

- $Q^*_t \overset{\text{def}}{=} Q^{\text{lin}}_{\theta_t}(\mathcal{X}^*) \in \mathbb{R}^{|\mathcal{X}^*| \times 1}$ – at new state-action pairs $\mathcal{X}^*$, using $\theta$ ($|\mathcal{X}^*|$ is the number of new state-action pairs we wish to evaluate).

## 4 THEORETICAL RESULTS

Here we present the results for the dynamics of neural network parameters during training under TD updates for a fixed policy and a static dataset of transitions. We consider two cases: first a TD update where gradients are propagated through all terms in the equation (no target network), and then the case with a target network through which no gradients are propagated, which is instead updated through soft updates of the parameters to converge to the primary value network (analogous to Haarnoja et al. (2018)). We will consider losses that minimize the dataset-averaged TD error throughout, which in the tabular setting guarantees convergence to the true value function in deterministic environments (although the same guarantee does not always hold for stochastic environments) (Sutton & Barto, 2018).

### 4.1 SINGLE Q-NETWORK

First, we consider the setup with no target network, where all gradients are passed through all terms in the TD update. While in practice this is rarely used due to instabilities in training, in principle the value function could be learned this way. This corresponds to the following loss in eq. 4:

$$\mathcal{L} = \frac{1}{2}\, \|Q_\theta(\mathcal{X}) - r - \gamma Q_\theta(\mathcal{X}')\|_2^2\,. \tag{6}$$

We note that the proofs by Lee et al. (2019) can be adapted to show that even in this setup the kernel is stable under gradient descent and the linearized approximation will also become arbitrarily exact with this update rule during training for wide enough networks. This follows immediately by replacing the original network $f$ in the relevant proofs with $Q - \gamma Q'$, resulting in the same proof with a different kernel for gradient descent. Hence, by substituting the linearization into the equation of motion and taking derivatives, we obtain

$$\dot{\omega}_t = -\eta \left( \nabla_\theta Q_0 - \gamma \nabla_\theta Q'_0 \right)^T \left( (Q_0 - \boldsymbol{r} - \gamma Q'_0) + (\nabla_\theta Q_0 - \gamma \nabla_\theta Q'_0) \omega_t \right). \tag{7}$$

Much like Lee et al. (2019), this is a linear differential equation in $\omega_t$ and can be solved analytically. We present the derivation in Appendix A, and report here the solution:

$$\omega_t = -\left( \nabla_\theta Q_0 - \gamma \nabla_\theta Q'_0 \right)^T \bar{\Theta}_0^{-1} \left( \boldsymbol{I}_{|\mathcal{D}|} - e^{-\eta \bar{\Theta}_0 t} \right) (Q_0 - \boldsymbol{r} - \gamma Q'_0), \tag{8}$$

where we introduce

$$\bar{\Theta}_0 \stackrel{\text{def}}{=} \left( \nabla_\theta Q_0 - \gamma \nabla_\theta Q'_0 \right) \left( \nabla_\theta Q_0 - \gamma \nabla_\theta Q'_0 \right)^T. \tag{9}$$

Similarly to Lee et al. (2019), the distribution of an ensemble of networks after training can also be computed in closed form, with a derivation also presented in Appendix A. The output $Q_{\theta_0}^{\text{lin}}(\mathcal{X}^*)$ has a distribution

$$\mathcal{N}\Big( Z\boldsymbol{r}, \mathcal{K}^{\mathcal{X}^* \mathcal{X}^*} - Z(\mathcal{K}^{\mathcal{X}\mathcal{X}^*} - \gamma \mathcal{K}^{\mathcal{X}' \mathcal{X}^*}) - \Big( Z(\mathcal{K}^{\mathcal{X}\mathcal{X}^*} - \gamma \mathcal{K}^{\mathcal{X}' \mathcal{X}^*}) \Big)^T$$
$$- Z(\mathcal{K}^{\mathcal{X}\mathcal{X}} - \gamma \mathcal{K}^{\mathcal{X}'\mathcal{X}} - \gamma \mathcal{K}^{\mathcal{X}\mathcal{X}'} + \gamma^2 \mathcal{K}^{\mathcal{X}'\mathcal{X}'}) Z^T \Big);$$
$$Z = \left( \Theta^{\mathcal{X}^* \mathcal{X}} - \gamma \Theta^{\mathcal{X}^* \mathcal{X}'} \right) \bar{\Theta}^{-1} \left( \boldsymbol{I}_{|\mathcal{D}|} - e^{-\eta \bar{\Theta} t} \right); \quad \bar{\Theta} = \Theta^{\mathcal{X}\mathcal{X}} - \gamma \Theta^{\mathcal{X}'\mathcal{X}} - \gamma \Theta^{\mathcal{X}\mathcal{X}'} + \gamma^2 \Theta^{\mathcal{X}'\mathcal{X}'}$$
$$\tag{10}$$

where $\mathcal{K}$ and $\Theta$ are the architecture-dependent NNGP kernels and NTK at initialization respectively (Lee et al., 2017; Jacot et al., 2018) which can be computed analytically for a family of architectures and activation functions.

Again, by analogy with the results by Lee et al. (2019), we note that the critical learning rate for approximating the continuous-time gradient descent dynamics is given by

$$\eta_{\text{critical}} = \frac{2}{\lambda_{\max} + \lambda_{\min}} \tag{11}$$

where $\lambda_{\max}$ and $\lambda_{\min}$ in this context are the maximum and minimum eigenvalues of $\Theta$. Finally, we remark that at every stage, setting $\gamma = 0$ reduces the equations to be identical to those by Lee et al. (2019), which is expected because in this regime learning the value function is essentially a supervised learning task of learning the immediate rewards, and all terms with policy dependence (those involving $\mathcal{X}'$) drop out as would be expected in a supervised learning task. We will consider cases with different values of $\gamma$.

## 4.2 WITH TARGET NETWORK

Here we consider a target network $\bar{\theta}$ which is updated through soft target network updates (Haarnoja et al., 2018). For discrete updates to the target network, the following equation is used:

$$\bar{\theta}_{t+1} = (1 - \tau)\bar{\theta}_t + \tau \theta_{t+1}, \tag{12}$$

where $\tau$ is the smoothing coefficient. The continuous-time equations of motion of the primary network parameters $\theta$ and the target network parameters $\bar{\theta}$ are then given by

$$\begin{cases} \dot{\omega}_t = -\eta \nabla_\theta Q_0^T \nabla_{Q_{\theta_t}^{\text{lin}}(\mathcal{X})} \mathcal{L} \\ \dot{\bar{\omega}}_t = \tau(\omega_t - \bar{\omega}_t), \end{cases} \tag{13}$$

where $\bar{\omega}_t = \bar{\theta}_t - \theta_0$ (we assume the target network is initialized with the same parameters $\bar{\theta}_0 = \theta_0$). In this case the loss is given by

$$\mathcal{L} = \frac{1}{2} \left\| \underbrace{Q_{\theta_t}^{\text{lin}}(\mathcal{X})}_{\text{primary network}} - \boldsymbol{r} - \gamma \underbrace{Q_{\bar{\theta}_t}^{\text{lin}}(\mathcal{X}')}_{\text{target network}} \right\|_2^2. \tag{14}$$

We note that the loss and udpate rules of the DQN are no longer purely involving gradient updates. Hence, the reasoning in sec. 4.1, justifying why reasoning analogous to Lee et al. (2019) guarantees the linearization approximation holds during training in the TD case, no longer directly applies. Nonetheless, we proceed with the analysis under the assumption it does. The loss presented here is much more common in practical RL algorithms, and we present our step-by-step analysis in Appendix B, and report here the solution:

$$\omega_t = \nabla_\theta Q_0^T \bar{\Theta}_0'^{-1} \left( \boldsymbol{E}_{11} - \boldsymbol{I}_{|\mathcal{D}|} + \tau^{-1} \eta \gamma \nabla_\theta Q_0' \nabla_\theta Q_0^T \boldsymbol{E}_{21} \right) \left( Q_0 - \boldsymbol{r} - \gamma Q_0' \right), \qquad (15)$$

where we introduce

$$\bar{\Theta}_0' \stackrel{\text{def}}{=} \left( \nabla_\theta Q_0 - \gamma \nabla_\theta Q_0' \right) \nabla_\theta Q_0^T, \qquad (16)$$

$\boldsymbol{E}_{11}$ and $\boldsymbol{E}_{21}$ are the block matrices from the result of the exponent:

$$\begin{bmatrix} \boldsymbol{E}_{11} & \boldsymbol{E}_{12} \\ \boldsymbol{E}_{21} & \boldsymbol{E}_{22} \end{bmatrix} = \exp \left( \begin{bmatrix} -\eta \nabla_\theta Q_0 \nabla_\theta Q_0^T & \eta \gamma \nabla_\theta Q_0' \nabla_\theta Q_0^T \\ \tau \boldsymbol{I}_{|\mathcal{D}|} & -\tau \boldsymbol{I}_{|\mathcal{D}|} \end{bmatrix} t \right). \qquad (17)$$

## 5 EXPERIMENTS

In this section, we present our experiments that show the described training dynamics. We consider a fully connected NN architecture, trained with full-batch gradient descent, using a sufficiently small learning rate (smaller than $\eta_{\text{critical}}$) so the approximation of continuous-time gradient descent holds well. We design a toy environment to better illustrate our theoretical results, and consider the cart pole environment as well. All experiments were developed in JAX (Bradbury et al., 2018), Haiku (Hennigan et al., 2020) and the Neural Tangents (Novak et al., 2020) libraries. We provide the code necessary to investigate RL training dynamics presented in this work.

### 5.1 TOY EXAMPLE

We consider a simple, interpretable toy environment to validate the analytic solutions we derived. In this environment, the state is represented by a single nonnegative number $s$. There is only one action available to the agent, which has the effect of increasing $s$ by $0.25$. When the action results in a new state that exceeds $s = 1$ a reward of $1$ is given, otherwise the reward is zero. An action from a state above $s = 1$ terminates the episode. This environment displays the two following key properties. First, it has a simple and interpretable ground truth value function: the value will be $1$ (taking $\gamma = 1$) for all $s < 1$ and $0$ otherwise. Moreover, inferring such a value will require the function approximator to correctly propagate cumulative rewards across multiple timesteps. These properties allow us to verify that the analytic solutions behave as expected and examine any discrepancies with the modeled ensemble of NNs.

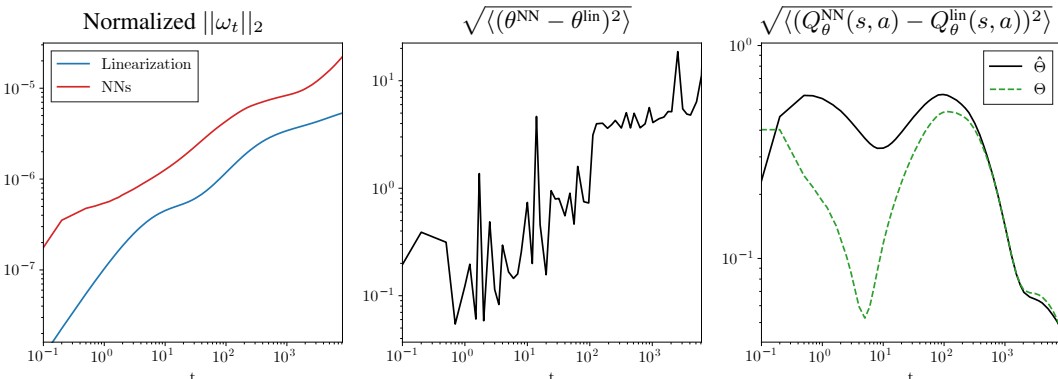

Figure 1: Training dynamics in the toy environment. The left pane shows how the parameters of the networks evolve from their starting point over time. The middle pane shows the RMSE between parameters obtained by training a NN and parameters from the analytic solution. The right pane shows how the output values of the Q-networks differ, comparing the Q-values of the linearization with the original neural network.

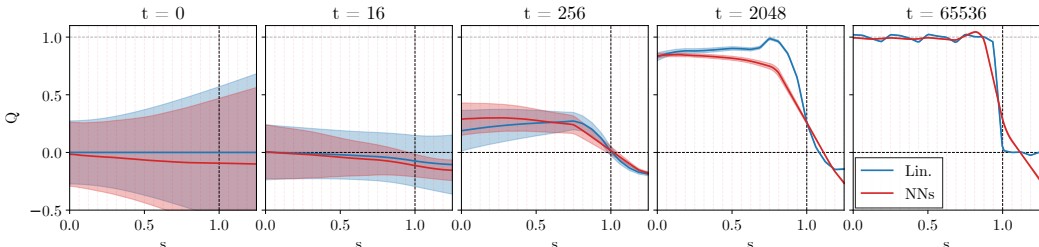

Figure 2: Q-network dynamics during training. The curves and shading correspond to mean and standard deviations of the output distributions at various stages during training. For individual run samples of the NN ensemble, see fig. 6 in the appendix. The dataset for this experiment is populated with transitions observed at the states with $s \in [0, 1.25]$ corresponding to the dashed vertical lines.

We apply our parameter dynamics solutions to this environment and compare them against empirically trained networks, with an architecture of 2 hidden layers that are wide but still computationally tractable (1024 neurons per layer). For the neural networks, we train each network for a specified amount of full-batch gradient steps, whereas in the analytic solutions we simply evaluate these at the desired $t$. Unless otherwise specified, the analytic quantities examined in this section are those derived from eq. 8. Further experiment details are presented in Appendix D.1.

First, we present a comparison of the evolution through time of an individual NN with a single initialization. In fig. 1, the left pane shows that the average distance traveled by the parameters during training is small, ensuring that we remain in the approximately linear regime. The middle and right panes show how the error between analytic prediction and trained NN evolves in parameter and function space. Interestingly, while error in parameter space tends to accumulate throughout training, as may be expected, in function space the error tends to alternate, eventually decreasing as both models tend to converge to the same function.

A more qualitatively intuitive visualization of the similarity between the analytic prediction (distribution given in eq. 10) and an ensemble of neural networks (50 different initializations) is presented in fig. 2. Qualitatively, we observe that the output from the analytic solution approximately matches the NNs' learning progression, suggesting that the approximation successfully models some of the core aspects of the learning.

During and after training, uncertainty quantification remains mostly consistent between the analytic solution and empirical NNs within the in-distribution region. However, in fig. 3 we see that the empirical NNs estimate smaller undertinies in certain regions of the out-of-distribution statespace. Moreover, within the training distribution, in fig. 4 we see that the exact analytic solution has smaller epistemic uncertainty at the exact points where data is available whereas the NN ensemble has a roughly constant, albeit small, uncertainty estimate throughout the in-distribution state-space. Hence, the uncertainty quantification provided by the NN ensemble in this case is suboptimal and inferior to the one provided by the linearized approximation.

We also compare experimentally the dynamics of NNs with target networks, using the linearized target network solution (eq. 15) and compare different values of $\tau$. These results can be seen in

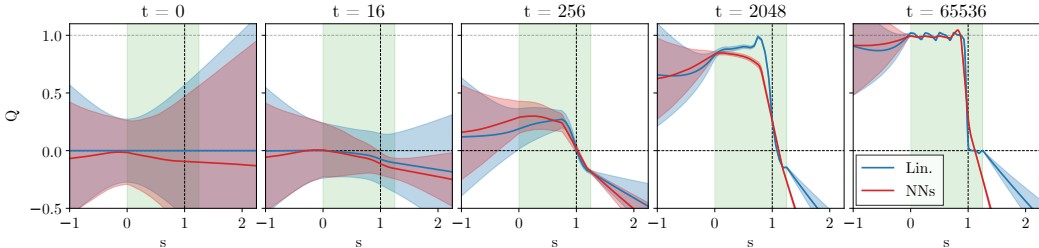

Figure 3: Q-network dynamics during training, with states outside the training distribution. Green area denotes states which were present in the training dataset.

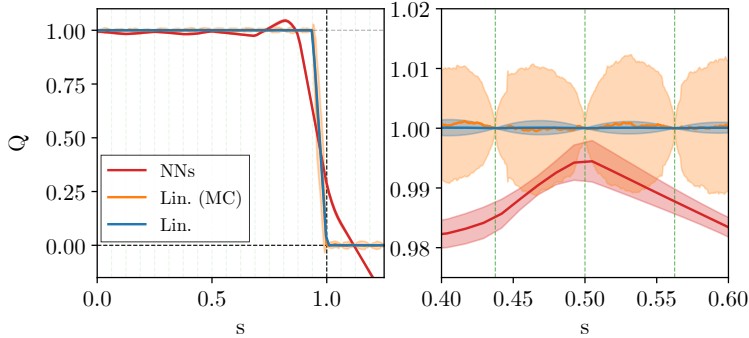

Figure 4: Comparison of our theoretical results to the empirical ensemble in the setup where a single Q-network estimates both Q-values. **Lin.** (blue) – eq. (10) applied to the toy environment ($t \to \infty$). **Lin. (MC)** (orange) – eq. (8, 5) evaluated with $500$ different intitializations. **NNs** (red) – an ensemble of $50$ independently trained Q-networks for $2^{16}$ full batch gradient descent steps. Vertical dashed green lines indicate the states $s$ from transitions present in the dataset $\mathcal{D}$. Vertical dashed black line at $s = 1$ indicates the threshold of the reward region. The networks have 2 hidden layers of width 1024, *ReLU* activation functions, $\sigma_w^2 = 1$, $\sigma_b^2 = 0.1$, and $\eta = 0.01$. The shaded regions denote 1 standard deviation from the mean. We observe that Lin. (MC) has much higher uncertainty than Lin., and note that this may be due to the closed-form prediction using infinite width.

fig. 7 in the Appendix D.1. We find empirically that the analytic solutions for the linearized target network do not match the true dynamics as closely as the single-network solutions do, suggesting that the linearized model does not provide good approximations to the dynamics in this regime.

## 5.2 CART POLE

In this section we consider the cart pole environment (Barto et al., 1983). Experiments in this subsection were implemented using the Gymnasium (Towers et al., 2023) package.

We train an ensemble of 50 fully-connected Q-networks each with two layers of width 256, with $\gamma = 1$. For the neural networks, we train each network for $2^{16}$ epochs. For the linearization, we consider eq. (8) (at $t = 2^{16}$). We evaluate 50 different initializations.

We then compare the Q-values obtained from the neural networks and the derived formulas against the true returns obtained by rolling out the evaluated policy in the environment to evaluate our method. Train and test refer to whether the policy is rolled out from states that are present in the dataset or not. Test and train states are sampled from the same initial state distribution. Further experiment details can be found in Appendix D.2. The results can be seen in table 1.

|  | Method | RMSE | NLL |
|---|---|---|---|
| Train | NNs | $231.2 \pm 0.3$ | $331625 \pm 201000$ |
|  | TD-NTK (ours) | $\mathbf{60.4 \pm 2.5}$ | $\mathbf{55.8 \pm 82.1}$ |
| Test | NNs | $196.8 \pm 0.3$ | $259429 \pm 134000$ |
|  | TD-NTK (ours) | $\mathbf{17.2 \pm 2.7}$ | $\mathbf{7.8 \pm 15.4}$ |

Table 1: Results from the cart pole environment (lower is better). **NNs** – an ensemble of independently trained Q-networks for $2^{16}$ full batch gradient descent steps. **TD-NTK** – analytic predictions from eq. (8) and (5) applied to the dataset. We evaluate the NN and linearization Q-values and compare them against the true returns, obtained from rolling out the policy from a given state. To compute the negative log-likelihood (NLL), we fit the Q-values from the ensemble to the Gaussian function, and use its probability density function.

These results again suggest that the linearization is able to better estimate the Q-values from the data. In particular, the error in predicted Q-values is significantly lower, as can be seen from the RMSE scores, and the uncertainty estimation is much better using the prediction from the linearized model, as can be seen from the NLL scores. We find these results surprising – we expected to see the

training dynamics of the linearization accurately describing the neural network training dynamics, but we instead observe that they better capture the epistemic uncertainty from the data.

## 6 CONCLUSION

In this work we have derived analytic closed-form solutions for the training dynamics of infinitely wide neural network ensembles trained under temporal difference updates with continuous-time gradient descent. We achieved this by using an extension of the NTK formalism to linearize the output of wide neural networks and solve the resulting first-order differential equations of motion for the parameters. Having established that the evolution through time of a single initialized network is an affine transformation of its output at initialization, we were able to derive analytical expressions for the evolution of an ensemble of randomly initialized, independently trained networks in terms of their NNGP kernels. This allows us to approximate the dynamics of real ensembles trained through gradient descent for policy evaluation in a reinforcement learning context, and predict the changes in uncertainty.

We have validated our solutions in a simple toy environment and have compared the result of carrying out our derived analytical policy evaluation against a real ensemble of neural networks on the classic control cart pole environment. Interestingly, we find that our analytic solution better predicts the returns and uncertainty of the policy being evaluated, with better predictive RMSE and better uncertainty demonstrated through improved NLL scores. This suggests that uncertainty quantification of real ensembles may suffer from unexpected behavior that degrades the quality of their predictive uncertainty in the TD setting.

We hope this work serves as a basis for analyzing TD training in deep RL, opening up the ability to study convergence and stability of TD algorithms with deep NNs. In addition, return uncertainty quantification for deep RL is a key open question in RL, and the methods provided here form a starting place for possible quantitative insight into the behavior of ensembles, a widespread heuristic method for uncertainty quantification (Smit et al., 2021; An et al., 2021; Osband et al., 2018). Our experimental results suggest that these can behave unexpectedly and suboptimally, and further research into this phenomenon may prove beneficial for a wide range of RL contexts.

**Limitations and future work.** In our derivations and experiments, we consider TD policy evaluation for a fixed policy. Another key part of RL algorithms is policy improvement, and a natural next step for extending this work is to investigate the modeling of a non-stationary, improving policy. Additionally, we did not provide a rigorous proof that the asymptotic behavior of wide NNs as linear systems holds during training with update rules other than gradient descent, such as with target network updates. In a similar vein, many of the other assumptions made here necessary for tractability (continuous-time gradient descent, full-batch training, infinite width, fixed policy and dataset) are often broken in practical RL algorithms. While we believe that our work still retains some of the crucial ingredients that make TD learning with deep NNs interesting, extending this work while relaxing some of these assumptions could lead to further valuable insight into other aspects of training practical RL agents.

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

# A  SINGLE Q-NETWORK DERIVATION

The dynamics governing gradient descent (GD) are well known and given as follows:

$$\theta_{t+1} = \theta_t - \eta \nabla_\theta \mathcal{L} \,. \tag{18}$$

Using continuous time GD, the dynamics can be written as

$$\dot{\theta}_t = -\eta \nabla_\theta \mathcal{L} \,. \tag{19}$$

Similar to Q-learning, we aim to minimize the Bellman error

$$\mathcal{L} = \| Q_\theta(\mathcal{X}) - \boldsymbol{r} - \gamma Q_\theta(\mathcal{X}') \|_2^2 \,, \tag{20}$$

and we note that in this equation, the two instances of $Q_\theta$ are using the exact same Q-network – this corresponds to the setup where we do not use a target network. We consider a linearized neural network in the same way as Lee et al. (2019), where we replace the Q-function with a linearized version, based on eq. (3):

$$Q_{\theta_t}^{\text{lin}}(\mathcal{X}) = Q_{\theta_0}^{\text{lin}}(\mathcal{X}) + \nabla_\theta Q_{\theta_0}^{\text{lin}}(\mathcal{X}) \, \omega_t \,, \tag{21}$$

where $\omega_t \overset{\text{def}}{=} \theta_t - \theta_0$, and $\dot{\omega}_t = \dot{\theta}_t$. For brevity we shorten the notation: $Q_0 \overset{\text{def}}{=} Q_{\theta_0}^{\text{lin}}(\mathcal{X})$, $Q_0' \overset{\text{def}}{=} Q_{\theta_0}^{\text{lin}}(\mathcal{X}')$. We present our derivations (with reference to Lee et al. (2019)) for the loss defined in eq. (20). Expanding eq. (19) yields

$$\begin{aligned}
\dot{\omega}_t &= -\eta \nabla_\theta Q_0^T \nabla_{Q_{\theta_0}^{\text{lin}}(\mathcal{X})} \mathcal{L} - \eta \nabla_\theta Q_0'^T \nabla_{Q_{\theta_0}^{\text{lin}}(\mathcal{X}')} \mathcal{L} \\
&= -\eta \left( \nabla_\theta Q_0 - \gamma \nabla_\theta Q_0' \right)^T \left( (Q_0 - \boldsymbol{r} - \gamma Q_0') + (\nabla_\theta Q_0 - \gamma \nabla_\theta Q_0') \, \omega_t \right) .
\end{aligned} \tag{22}$$

Therefore, we obtain

$$\omega_t = -\left( \nabla_\theta Q_0 - \gamma \nabla_\theta Q_0' \right)^T \bar{\Theta}_0^{-1} \left( \boldsymbol{I}_{|\mathcal{D}|} - e^{-\eta \bar{\Theta}_0 t} \right) (Q_0 - \boldsymbol{r} - \gamma Q_0') \,, \tag{23}$$

where $\bar{\Theta}_0 \overset{\text{def}}{=} \left( \nabla_\theta Q_0 - \gamma \nabla_\theta Q_0' \right) \left( \nabla_\theta Q_0 - \gamma \nabla_\theta Q_0' \right)^T$. Eq. (21) now becomes:

$$Q_{\theta_t}^{\text{lin}}(\mathcal{X}^*) = Q_0^* - \nabla_\theta Q_0^* \left( \nabla_\theta Q_0 - \gamma \nabla_\theta Q_0' \right)^T \bar{\Theta}_0^{-1} \left( \boldsymbol{I}_{|\mathcal{D}|} - e^{-\eta \bar{\Theta}_0 t} \right) (Q_0 - \boldsymbol{r} - \gamma Q_0') \,, \tag{24}$$

where $Q_0^* \overset{\text{def}}{=} Q_{\theta_0}^{\text{lin}}(\mathcal{X}^*)$. We can separate the equation into two terms:

$$Q_{\theta_t}^{\text{lin}}(\mathcal{X}^*) = \mu(\mathcal{X}^*) + \varsigma(\mathcal{X}^*) \,, \tag{25}$$

where

$$\mu(\mathcal{X}^*) = \nabla_\theta Q_0^* \left( \nabla_\theta Q_0 - \gamma \nabla_\theta Q_0' \right)^T \bar{\Theta}_0^{-1} \left( \boldsymbol{I}_{|\mathcal{D}|} - e^{-\eta \bar{\Theta}_0 t} \right) \boldsymbol{r} \,, \tag{26}$$

$$\varsigma(\mathcal{X}^*) = Q_0^* - \nabla_\theta Q_0^* \left( \nabla_\theta Q_0 - \gamma \nabla_\theta Q_0' \right)^T \bar{\Theta}_0^{-1} \left( \boldsymbol{I}_{|\mathcal{D}|} - e^{-\eta \bar{\Theta}_0 t} \right) (Q_0 - \gamma Q_0') \tag{27}$$

Taking the expectation of these terms yields

$$\mathbb{E}[Q_{\theta_t}^{\text{lin}}(\mathcal{X}^*)] = (\Theta^{\mathcal{X}^* \mathcal{X}} - \gamma \Theta^{\mathcal{X}^* \mathcal{X}'}) \bar{\Theta}^{-1} \left( \boldsymbol{I}_{|\mathcal{D}|} - e^{-\eta \bar{\Theta} t} \right) \boldsymbol{r} \,, \tag{28}$$

$$\begin{aligned}
\mathbb{V}[Q_{\theta_t}^{\text{lin}}(\mathcal{X}^*)] = {} & \mathcal{K}^{\mathcal{X}^* \mathcal{X}^*} - (\mathcal{K}^{\mathcal{X}^* \mathcal{X}} - \gamma \mathcal{K}^{\mathcal{X}^* \mathcal{X}'}) \left( \boldsymbol{I}_{|\mathcal{D}|} - e^{-\eta \bar{\Theta} t} \right) \bar{\Theta}^{-1} (\Theta^{\mathcal{X} \mathcal{X}^*} - \gamma \Theta^{\mathcal{X}' \mathcal{X}^*}) \\
& - (\Theta^{\mathcal{X}^* \mathcal{X}} - \gamma \Theta^{\mathcal{X}^* \mathcal{X}'}) \bar{\Theta}^{-1} \left( \boldsymbol{I}_{|\mathcal{D}|} - e^{-\eta \bar{\Theta} t} \right) (\mathcal{K}^{\mathcal{X} \mathcal{X}^*} - \gamma \mathcal{K}^{\mathcal{X}' \mathcal{X}^*}) \\
& - (\Theta^{\mathcal{X}^* \mathcal{X}} - \gamma \Theta^{\mathcal{X}^* \mathcal{X}'}) \bar{\Theta}^{-1} \left( \boldsymbol{I}_{|\mathcal{D}|} - e^{-\eta \bar{\Theta} t} \right) (\mathcal{K}^{\mathcal{X} \mathcal{X}} - \gamma \mathcal{K}^{\mathcal{X}' \mathcal{X}} - \gamma \mathcal{K}^{\mathcal{X} \mathcal{X}'} + \gamma^2 \mathcal{K}^{\mathcal{X}' \mathcal{X}'}) \\
& \left( \boldsymbol{I}_{|\mathcal{D}|} - e^{-\eta \bar{\Theta} t} \right) \bar{\Theta}^{-1} (\Theta^{\mathcal{X} \mathcal{X}^*} - \gamma \Theta^{\mathcal{X}' \mathcal{X}^*})
\end{aligned} \tag{29}$$

where $\mathcal{K}^{\mathcal{X}' \mathcal{X}} = \mathbb{E}[Q_0' Q_0^T]$, and similarly for other combinations,

$$\bar{\Theta} \overset{\text{def}}{=} \Theta^{\mathcal{X} \mathcal{X}} - \gamma \Theta^{\mathcal{X}' \mathcal{X}} - \gamma \Theta^{\mathcal{X} \mathcal{X}'} + \gamma^2 \Theta^{\mathcal{X}' \mathcal{X}'} \,, \tag{30}$$

where $\Theta^{\mathcal{X}' \mathcal{X}} = \mathbb{E}[\nabla_\theta Q_0' \nabla_\theta Q_0^T]$, and similarly for other combinations.

We use these expressions in the environments, see sec. 5.

At the point where the network converges, i.e., at $t \to \infty$, eq. (24) becomes:

$$Q_{\theta_\infty}^{\text{lin}}(\mathcal{X}^*) = Q_0^* - \nabla_\theta Q_0^* \left( \nabla_\theta Q_0 - \gamma \nabla_\theta Q_0' \right)^T \bar{\Theta}_0^{-1} (Q_0 - \boldsymbol{r} - \gamma Q_0') \,. \tag{31}$$

**Case $\gamma = 0$.** To check consistency with the supervised learning case, we verify what happens to the system when $\gamma = 0$:

$$Q_{\theta_\infty}^{\text{lin}}(\mathcal{X}^*) = Q_0^* - \nabla_\theta Q_0^* \nabla_\theta Q_0^T \left(\nabla_\theta Q_0 \nabla_\theta Q_0^T\right)^{-1}(Q_0 - \boldsymbol{r}) , \tag{32}$$

and evaluated at the same training points, i.e., when $\mathcal{X}^* = \mathcal{X}$ the equation reduces to:

$$Q_{\theta_\infty}^{\text{lin}}(\mathcal{X}) = \boldsymbol{r} . \tag{33}$$

As expected, the neural network converges to the reward after training. This corresponds to the supervised learning case, where we can replace the predictions $Q(\mathcal{X})$ with $\boldsymbol{f}(\mathcal{X})$ and the target values $\boldsymbol{r}$ with $\boldsymbol{y}$.

## B  WITH TARGET Q-NETWORK DERIVATION

Here we consider having a target network that uses its own parameters $\bar{\theta}$, but otherwise acts like the primary Q-network. The target network parameters are set using a smooth moving average:

$$\bar{\theta}_{t+1} = (1 - \tau)\bar{\theta}_t + \tau\theta_{t+1} , \tag{34}$$

where $\tau$ is the smoothing coefficient.

We consider the Bellman error with two Q-networks – a primary network and a target network:

$$\mathcal{L} = \frac{1}{2} \left\| Q_{\underset{\text{primary network}}{\theta_t}}^{\text{lin}}(\mathcal{X}) - \boldsymbol{r} - \gamma Q_{\underset{\text{target network}}{\bar{\theta}_t}}^{\text{lin}}(\mathcal{X}') \right\|_2^2 . \tag{35}$$

Here we present our derivation for this setting as well. The dynamics of the main Q-network are governed by the same equation (19) as in the previous section, except additionally the target network is updated as a smooth moving average of the main network:

$$\begin{cases} \dot{\omega}_t = -\eta \nabla_\theta Q_0^T \nabla_{Q_{\theta_t}^{\text{lin}}(\mathcal{X})} \mathcal{L} \\ \dot{\bar{\omega}}_t = \tau(\omega_t - \bar{\omega}_t) , \end{cases} \tag{36}$$

and expanding this yields

$$\begin{cases} \dot{\omega}_t = -\eta \left(\nabla_\theta Q_0 - \gamma \nabla_\theta Q_0'\right)^T \left((Q_0 - \boldsymbol{r} - \gamma Q_0') + \nabla_\theta Q_0 \, \omega_t - \gamma \nabla_\theta Q_0' \, \bar{\omega}_t\right) , \\ \dot{\bar{\omega}}_t = \tau(\omega_t - \bar{\omega}_t) , \end{cases} \tag{37}$$

where $\bar{\omega}_t \overset{\text{def}}{=} \bar{\theta}_t - \theta_0$ (we also assume $\bar{\theta}_0 = \theta_0$, implying the two networks are initialized with the same parameters); we use the same shorthand notation for Q-values as in the previous section, except computing the second Q-value in the equation using the target network $Q_0' \overset{\text{def}}{=} Q_{\bar{\theta}_0}^{\text{lin}}(\mathcal{X}')$.

We can rewrite the system (37) in matrix form:

$$\begin{bmatrix} \dot{\omega}_t \\ \dot{\bar{\omega}}_t \end{bmatrix} = \begin{bmatrix} -\eta \boldsymbol{U}^T \boldsymbol{U} & \eta\gamma \boldsymbol{U}^T \boldsymbol{V} \\ \tau \boldsymbol{I}_{|\theta|} & -\tau \boldsymbol{I}_{|\theta|} \end{bmatrix} \begin{bmatrix} \omega_t \\ \bar{\omega}_t \end{bmatrix} + \begin{bmatrix} -\eta \boldsymbol{U}^T \hat{\boldsymbol{r}} \\ 0 \end{bmatrix} , \tag{38}$$

where $\boldsymbol{U} \overset{\text{def}}{=} \nabla_\theta Q_0$, $\boldsymbol{V} \overset{\text{def}}{=} \nabla_\theta Q_0'$, $\hat{\boldsymbol{r}} \overset{\text{def}}{=} Q_0 - \boldsymbol{r} - \gamma Q_0'$.

Therefore, the solution is

$$\begin{bmatrix} \omega_t \\ \bar{\omega}_t \end{bmatrix} = \begin{bmatrix} -\eta \boldsymbol{U}^T \boldsymbol{U} & \eta\gamma \boldsymbol{U}^T \boldsymbol{V} \\ \tau \boldsymbol{I}_{|\theta|} & -\tau \boldsymbol{I}_{|\theta|} \end{bmatrix}^{-1} \left(\exp\left(\begin{bmatrix} -\eta \boldsymbol{U}^T \boldsymbol{U} & \eta\gamma \boldsymbol{U}^T \boldsymbol{V} \\ \tau \boldsymbol{I}_{|\theta|} & -\tau \boldsymbol{I}_{|\theta|} \end{bmatrix} t\right) - \boldsymbol{I}_{2|\theta|}\right) \begin{bmatrix} -\eta \boldsymbol{U}^T \hat{\boldsymbol{r}} \\ 0 \end{bmatrix} . \tag{39}$$

and at $t \to \infty$ it becomes

$$\omega_t = -\nabla_\theta Q_0^T \bar{\Theta}_0'^{-1}(Q_0 - \boldsymbol{r} - \gamma Q_0') , \tag{40}$$

where $\bar{\Theta}_0' = \left(\nabla_\theta Q_0 - \gamma \nabla_\theta Q_0'\right)\nabla_\theta Q_0^T$. For a new state-action pair $\mathcal{X}^*$ the Q-function is:

$$Q_{\theta_\infty}^{\text{lin}}(\mathcal{X}^*) = Q_0^* - \nabla_\theta Q_0^* \nabla_\theta Q_0^T \bar{\Theta}_0'^{-1} (Q_0 - \boldsymbol{r} - \gamma Q_0') \ . \tag{41}$$

Rewriting eq. (39) (details in sec. C.3), we obtain:

$$\begin{bmatrix} \omega_t \\ \bar{\omega}_t \end{bmatrix} = \begin{bmatrix} \boldsymbol{U}^T & 0 \\ 0 & \boldsymbol{U}^T \end{bmatrix} \begin{bmatrix} -\eta \boldsymbol{U}\boldsymbol{U}^T & \eta\gamma \boldsymbol{V}\boldsymbol{U}^T \\ \tau \boldsymbol{I}_{|\mathcal{D}|} & -\tau \boldsymbol{I}_{|\mathcal{D}|} \end{bmatrix}^{-1} \begin{bmatrix} -\eta \left( \boldsymbol{E}_{11} - \boldsymbol{I}_{|\mathcal{D}|} \right) \hat{\boldsymbol{r}} \\ -\eta \boldsymbol{E}_{21} \hat{\boldsymbol{r}} \end{bmatrix} , \tag{42}$$

or, for $\omega_t$, simply

$$\omega_t = \boldsymbol{U}^T \left( \boldsymbol{U}\boldsymbol{U}^T - \gamma \boldsymbol{V}\boldsymbol{U}^T \right)^{-1} \left( \boldsymbol{E}_{11} - \boldsymbol{I}_{|\mathcal{D}|} + \tau^{-1}\eta\gamma \boldsymbol{V}\boldsymbol{U}^T \boldsymbol{E}_{21} \right) \hat{\boldsymbol{r}} , \tag{43}$$

where $\boldsymbol{E}_{11}$ and $\boldsymbol{E}_{21}$ are the block matrices from the result of the exponent:

$$\begin{bmatrix} \boldsymbol{E}_{11} & \boldsymbol{E}_{12} \\ \boldsymbol{E}_{21} & \boldsymbol{E}_{22} \end{bmatrix} = \exp \left( \begin{bmatrix} -\eta \boldsymbol{U}\boldsymbol{U}^T & \eta\gamma \boldsymbol{V}\boldsymbol{U}^T \\ \tau \boldsymbol{I}_{|\mathcal{D}|} & -\tau \boldsymbol{I}_{|\mathcal{D}|} \end{bmatrix} t \right) . \tag{44}$$

We use this form to study what happens to the dynamics during training (see fig. 2). We now analyze certain cases of $\gamma$ and $\tau$.

**Case $\gamma = 0$.** Again, to check consistency with the supervised learning case, we verify what happens to the system when $\gamma = 0$:

$$\begin{bmatrix} \omega_t \\ \bar{\omega}_t \end{bmatrix} = \begin{bmatrix} \boldsymbol{U}^T & 0 \\ 0 & \boldsymbol{U}^T \end{bmatrix} \begin{bmatrix} -\eta^{-1}(\boldsymbol{U}\boldsymbol{U}^T)^{-1} & 0 \\ -\eta^{-1}(\boldsymbol{U}\boldsymbol{U}^T)^{-1} & -\tau^{-1}\boldsymbol{I}_{|\mathcal{D}|} \end{bmatrix} \begin{bmatrix} -\eta \left( \boldsymbol{E}_{11} - \boldsymbol{I}_{|\mathcal{D}|} \right) \hat{\boldsymbol{r}} \\ -\eta \boldsymbol{E}_{21} \hat{\boldsymbol{r}} \end{bmatrix} . \tag{45}$$

At $t \to \infty$ terms $\boldsymbol{E}_{11}$ and $\boldsymbol{E}_{21}$ tend to 0, and the dynamics simplify considerably

$$\omega_t = -\nabla_\theta Q_0^T (\nabla_\theta Q_0 \nabla_\theta Q_0^T)^{-1}(Q_0 - \boldsymbol{r}) , \tag{46}$$

giving the same result as in eq. (32), and the same analysis can be made as in eq. (33), i.e., the prediction converges to the reward, $Q_{\theta_\infty}^{\text{lin}}(\mathcal{X}) = \boldsymbol{r}$.

**Case $\tau$ is negative.** To verify the robustness of the equations, we check the case when $\tau$ is negative and hope to expect divergence. The diagonal in the exponent (44) is no longer negative semidefinite. This may cause divergence at cases when $t \to \infty$. Here we do not study the convergence properties of this setting, although there has been some work looking into convergence bounds via eigenvalues of the neural tangent kernel (Nguyen et al., 2021).

**Case $\tau = 0$.** In this case we can simulate the setting, where a hard update to the target network parameters occurs (Mnih et al., 2015); $\tau = 0$ would describe the dynamics between these updates. Our weight dynamics simplify to:

$$\begin{cases} \dot{\omega}_t = -\eta \nabla_\theta Q_0^T \, \nabla_{Q_{\theta_t}^{\text{lin}}(\mathcal{X})} \mathcal{L} \\ \dot{\bar{\omega}}_t = 0 \end{cases} \tag{47}$$

reducing the system to the case where the target network is fixed. The solution is the same as (43), except the term with $\tau^{-1}$ does not exist.

**Mean & covariance.** We can separate the equation into two terms:

$$Q_{\theta_t}^{\text{lin}}(\mathcal{X}^*) = \mu(\mathcal{X}^*) + \varsigma(\mathcal{X}^*) , \tag{48}$$

where

$$\mu(\mathcal{X}^*) = \nabla_\theta Q_0^* \boldsymbol{U}^T \left( \boldsymbol{U}\boldsymbol{U}^T - \gamma \boldsymbol{V}\boldsymbol{U}^T \right)^{-1} \left( \boldsymbol{I}_{|\mathcal{D}|} - \boldsymbol{E}_{11} - \tau^{-1}\eta\gamma \boldsymbol{V}\boldsymbol{U}^T \boldsymbol{E}_{21} \right) \boldsymbol{r} , \tag{49}$$

$$\varsigma(\mathcal{X}^*) = Q_0^* - \nabla_\theta Q_0^* \boldsymbol{U}^T \left( \boldsymbol{U}\boldsymbol{U}^T - \gamma \boldsymbol{V}\boldsymbol{U}^T \right)^{-1} \left( \boldsymbol{I}_{|\mathcal{D}|} - \boldsymbol{E}_{11} - \tau^{-1}\eta\gamma \boldsymbol{V}\boldsymbol{U}^T \boldsymbol{E}_{21} \right) (Q_0 - \gamma Q_0') , \tag{50}$$

The expectation of these terms is

$$\mathbb{E}[Q_{\theta_t}^{\text{lin}}(\mathcal{X}^*)] = \mu(\mathcal{X}^*) , \tag{51}$$

$$\begin{aligned} \mathbb{V}[Q_{\theta_t}^{\text{lin}}(\mathcal{X}^*)] &= \mathcal{K}^{\mathcal{X}^*\mathcal{X}^*} - (Z(\mathcal{K}^{\mathcal{X}\mathcal{X}^*} - \gamma\mathcal{K}^{\mathcal{X}'\mathcal{X}^*}))^T - Z(\mathcal{K}^{\mathcal{X}\mathcal{X}^*} - \gamma\mathcal{K}^{\mathcal{X}'\mathcal{X}^*}) \\ &\quad + Z(\mathcal{K}^{\mathcal{X}\mathcal{X}} - \gamma\mathcal{K}^{\mathcal{X}\mathcal{X}'} - \gamma\mathcal{K}^{\mathcal{X}'\mathcal{X}} + \gamma^2\mathcal{K}^{\mathcal{X}'\mathcal{X}'})Z^T , \end{aligned} \tag{52}$$

where $Z = \Theta^{\mathcal{X}^*\mathcal{X}} \left( \Theta^{\mathcal{X}\mathcal{X}} - \gamma\Theta^{\mathcal{X}'\mathcal{X}} \right)^{-1} \left( \boldsymbol{I}_{|\mathcal{D}|} - \boldsymbol{E}_{11} - \tau^{-1}\eta\gamma\Theta^{\mathcal{X}'\mathcal{X}} \boldsymbol{E}_{21} \right)$.

## C    OTHER PROOFS & DERIVATIONS

### C.1    SINGLE Q-NETWORK DIFFERENTIAL EQUATION

> For a differential equation of the form
>
> $$\dot{\boldsymbol{f}}_t = -\alpha \boldsymbol{D}^T \left( \boldsymbol{b} + \boldsymbol{D}\boldsymbol{f}_t \right), \tag{53}$$
>
> the solution is given by
>
> $$\boldsymbol{f}_t = -\boldsymbol{D}^T (\boldsymbol{D}\boldsymbol{D}^T)^{-1} \left( \boldsymbol{b} - e^{-\alpha \boldsymbol{D}\boldsymbol{D}^T t} \boldsymbol{c} \right), \tag{54}$$
>
> where $\boldsymbol{c}$ is a free variable. Additionally, if $\boldsymbol{f}_0 = 0$, then $\boldsymbol{c} = \boldsymbol{b}$.

We will present a straightforward proof by verification. When $\boldsymbol{f}_0 = 0$, we can check the point $t = 0$ to obtain the value for $\boldsymbol{c}$:

$$\boldsymbol{f}_0 = -\boldsymbol{D}^T (\boldsymbol{D}\boldsymbol{D}^T)^{-1} \left( \boldsymbol{b} - e^{\boldsymbol{0}} \boldsymbol{c} \right) \tag{55}$$

$$\boldsymbol{0} = -\boldsymbol{D}^T (\boldsymbol{D}\boldsymbol{D}^T)^{-1} \left( \boldsymbol{b} - \boldsymbol{c} \right), \tag{56}$$

the only general solution for $\boldsymbol{c}$ is $\boldsymbol{c} = \boldsymbol{b}$.

The derivative of eq. (54) is given by

$$
\begin{aligned}
\dot{\boldsymbol{f}}_t &= -\boldsymbol{D}^T (\boldsymbol{D}\boldsymbol{D}^T)^{-1} \left( -\left( -\alpha \boldsymbol{D}\boldsymbol{D}^T \right) e^{-\alpha \boldsymbol{D}\boldsymbol{D}^T t} \boldsymbol{c} \right) \\
&= -\alpha \boldsymbol{D}^T (\boldsymbol{D}\boldsymbol{D}^T)^{-1} \boldsymbol{D}\boldsymbol{D}^T e^{-\alpha \boldsymbol{D}\boldsymbol{D}^T t} \boldsymbol{c} \\
&= -\alpha \boldsymbol{D}^T e^{-\alpha \boldsymbol{D}\boldsymbol{D}^T t} \boldsymbol{c} \\
&= -\alpha \boldsymbol{D}^T \left( \boldsymbol{b} + e^{-\alpha \boldsymbol{D}\boldsymbol{D}^T t} \boldsymbol{c} - \boldsymbol{b} \right) \\
&= -\alpha \boldsymbol{D}^T \left( \boldsymbol{b} + \boldsymbol{D}\boldsymbol{D}^T (\boldsymbol{D}\boldsymbol{D}^T)^{-1} \left( e^{-\alpha \boldsymbol{D}\boldsymbol{D}^T t} \boldsymbol{c} - \boldsymbol{b} \right) \right) \\
&= -\alpha \boldsymbol{D}^T \left( \boldsymbol{b} + \boldsymbol{D} \left( -\boldsymbol{D}^T (\boldsymbol{D}\boldsymbol{D}^T)^{-1} \left( \boldsymbol{b} - e^{-\alpha \boldsymbol{D}\boldsymbol{D}^T t} \boldsymbol{c} \right) \right) \right) \\
&= -\alpha \boldsymbol{D}^T \left( \boldsymbol{b} + \boldsymbol{D}\boldsymbol{f}_t \right),
\end{aligned}
\tag{57}
$$

which is equal to the original differential eq. (53).

$\square$

### C.2    WITH TARGET Q-NETWORK DIFFERENTIAL EQUATION

> For a differential equation of the form
>
> $$\begin{bmatrix} \dot{\boldsymbol{f}}_t \\ \dot{\boldsymbol{g}}_t \end{bmatrix} = \begin{bmatrix} \boldsymbol{A} & \boldsymbol{B} \\ \boldsymbol{C} & \boldsymbol{D} \end{bmatrix} \begin{bmatrix} \boldsymbol{f}_t \\ \boldsymbol{g}_t \end{bmatrix} + \begin{bmatrix} \boldsymbol{e} \\ \boldsymbol{h} \end{bmatrix}, \tag{58}$$
>
> when $\boldsymbol{f}_0 = 0$ and $\boldsymbol{g}_0 = 0$ the solution is given by:
>
> $$\begin{bmatrix} \boldsymbol{f}_t \\ \boldsymbol{g}_t \end{bmatrix} = \begin{bmatrix} \boldsymbol{A} & \boldsymbol{B} \\ \boldsymbol{C} & \boldsymbol{D} \end{bmatrix}^{-1} \left( \exp \left( \begin{bmatrix} \boldsymbol{A} & \boldsymbol{B} \\ \boldsymbol{C} & \boldsymbol{D} \end{bmatrix} t \right) - \begin{bmatrix} \boldsymbol{I} & 0 \\ 0 & \boldsymbol{I} \end{bmatrix} \right) \begin{bmatrix} \boldsymbol{e} \\ \boldsymbol{h} \end{bmatrix}. \tag{59}$$
>
> The expression for the inverse can be obtained using the known block inverse formula (Bernstein, 2009):
>
> $$\begin{bmatrix} \boldsymbol{A} & \boldsymbol{B} \\ \boldsymbol{C} & \boldsymbol{D} \end{bmatrix}^{-1} = \begin{bmatrix} \left( \boldsymbol{A} - \boldsymbol{B}\boldsymbol{D}^{-1}\boldsymbol{C} \right)^{-1} & -\left( \boldsymbol{A} - \boldsymbol{B}\boldsymbol{D}^{-1}\boldsymbol{C} \right)^{-1} \boldsymbol{B}\boldsymbol{D}^{-1} \\ -\boldsymbol{D}^{-1}\boldsymbol{C} \left( \boldsymbol{A} - \boldsymbol{B}\boldsymbol{D}^{-1}\boldsymbol{C} \right)^{-1} & \boldsymbol{D}^{-1} + \boldsymbol{D}^{-1}\boldsymbol{C} \left( \boldsymbol{A} - \boldsymbol{B}\boldsymbol{D}^{-1}\boldsymbol{C} \right)^{-1} \boldsymbol{B}\boldsymbol{D}^{-1} \end{bmatrix}$$
>
> We present a proof in sec. C.2.

In essence, this is the same differential equation as (53), although in a matrix form. We present a proof by verification. The derivative of (59) is

$$
\begin{aligned}
\begin{bmatrix} \dot{\boldsymbol{f}}_t \\ \dot{\boldsymbol{g}}_t \end{bmatrix} &= \begin{bmatrix} \boldsymbol{A} & \boldsymbol{B} \\ \boldsymbol{C} & \boldsymbol{D} \end{bmatrix}^{-1} \begin{bmatrix} \boldsymbol{A} & \boldsymbol{B} \\ \boldsymbol{C} & \boldsymbol{D} \end{bmatrix} \exp\left( \begin{bmatrix} \boldsymbol{A} & \boldsymbol{B} \\ \boldsymbol{C} & \boldsymbol{D} \end{bmatrix} t \right) \begin{bmatrix} \boldsymbol{e} \\ \boldsymbol{h} \end{bmatrix} \\
&= \begin{bmatrix} \boldsymbol{A} & \boldsymbol{B} \\ \boldsymbol{C} & \boldsymbol{D} \end{bmatrix} \begin{bmatrix} \boldsymbol{A} & \boldsymbol{B} \\ \boldsymbol{C} & \boldsymbol{D} \end{bmatrix}^{-1} \left( \exp\left( \begin{bmatrix} \boldsymbol{A} & \boldsymbol{B} \\ \boldsymbol{C} & \boldsymbol{D} \end{bmatrix} t \right) - \begin{bmatrix} \boldsymbol{I} & 0 \\ 0 & \boldsymbol{I} \end{bmatrix} + \begin{bmatrix} \boldsymbol{I} & 0 \\ 0 & \boldsymbol{I} \end{bmatrix} \right) \begin{bmatrix} \boldsymbol{e} \\ \boldsymbol{h} \end{bmatrix} \\
&= \begin{bmatrix} \boldsymbol{A} & \boldsymbol{B} \\ \boldsymbol{C} & \boldsymbol{D} \end{bmatrix} \left( \begin{bmatrix} \boldsymbol{A} & \boldsymbol{B} \\ \boldsymbol{C} & \boldsymbol{D} \end{bmatrix}^{-1} \left( \exp\left( \begin{bmatrix} \boldsymbol{A} & \boldsymbol{B} \\ \boldsymbol{C} & \boldsymbol{D} \end{bmatrix} t \right) - \begin{bmatrix} \boldsymbol{I} & 0 \\ 0 & \boldsymbol{I} \end{bmatrix} \right) \begin{bmatrix} \boldsymbol{e} \\ \boldsymbol{h} \end{bmatrix} \right) + \begin{bmatrix} \boldsymbol{e} \\ \boldsymbol{h} \end{bmatrix} \\
&= \begin{bmatrix} \boldsymbol{A} & \boldsymbol{B} \\ \boldsymbol{C} & \boldsymbol{D} \end{bmatrix} \begin{bmatrix} \boldsymbol{f}_t \\ \boldsymbol{g}_t \end{bmatrix} + \begin{bmatrix} \boldsymbol{e} \\ \boldsymbol{h} \end{bmatrix} ,
\end{aligned}
\tag{60}
$$

which is equal to the original differential eq. (58).

$\square$

### C.3 Matrix Push-Through Derivation With Target Network

We begin with eq. (39):

$$
\begin{bmatrix} \omega_t \\ \bar{\omega}_t \end{bmatrix} = \begin{bmatrix} -\eta \boldsymbol{U}^T \boldsymbol{U} & \eta\gamma \boldsymbol{U}^T \boldsymbol{V} \\ \tau \boldsymbol{I}_{|\theta|} & -\tau \boldsymbol{I}_{|\theta|} \end{bmatrix}^{-1} \left( \exp\left( \begin{bmatrix} -\eta \boldsymbol{U}^T \boldsymbol{U} & \eta\gamma \boldsymbol{U}^T \boldsymbol{V} \\ \tau \boldsymbol{I}_{|\theta|} & -\tau \boldsymbol{I}_{|\theta|} \end{bmatrix} t \right) - \boldsymbol{I}_{2|\theta|} \right) \begin{bmatrix} -\eta \boldsymbol{U}^T \hat{\boldsymbol{r}} \\ 0 \end{bmatrix} . \tag{61}
$$

Rewriting the terms:

$$
\begin{aligned}
&\begin{bmatrix} -\eta \boldsymbol{U}^T \boldsymbol{U} & \eta\gamma \boldsymbol{U}^T \boldsymbol{V} \\ \tau \boldsymbol{I}_{|\theta|} & -\tau \boldsymbol{I}_{|\theta|} \end{bmatrix}^{-1} \left( \exp\left( \begin{bmatrix} -\eta \boldsymbol{U}^T \boldsymbol{U} & \eta\gamma \boldsymbol{U}^T \boldsymbol{V} \\ \tau \boldsymbol{I}_{|\theta|} & -\tau \boldsymbol{I}_{|\theta|} \end{bmatrix} t \right) - \boldsymbol{I}_{2|\theta|} \right) \begin{bmatrix} -\eta \boldsymbol{U}^T \hat{\boldsymbol{r}} \\ 0 \end{bmatrix} \\
&= \begin{bmatrix} -\eta \boldsymbol{U}^T \boldsymbol{U} & \eta\gamma \boldsymbol{U}^T \boldsymbol{V} \\ \tau \boldsymbol{I}_{|\theta|} & -\tau \boldsymbol{I}_{|\theta|} \end{bmatrix}^{-1} \begin{bmatrix} -\eta \boldsymbol{U}^T \left( \boldsymbol{E}_{11} - \boldsymbol{I}_{|\mathcal{D}|} \right) \hat{\boldsymbol{r}} \\ -\eta \boldsymbol{U}^T \boldsymbol{E}_{21} \hat{\boldsymbol{r}} \end{bmatrix} \\
&= \begin{bmatrix} -\eta \boldsymbol{U}^T \boldsymbol{U} & \eta\gamma \boldsymbol{U}^T \boldsymbol{V} \\ \tau \boldsymbol{I}_{|\theta|} & -\tau \boldsymbol{I}_{|\theta|} \end{bmatrix}^{-1} \begin{bmatrix} \boldsymbol{U}^T & 0 \\ 0 & \boldsymbol{U}^T \end{bmatrix} \begin{bmatrix} -\eta \left( \boldsymbol{E}_{11} - \boldsymbol{I}_{|\mathcal{D}|} \right) \hat{\boldsymbol{r}} \\ -\eta \boldsymbol{E}_{21} \hat{\boldsymbol{r}} \end{bmatrix}
\end{aligned}
\tag{62}
$$

where $\boldsymbol{E}_{11}$ and $\boldsymbol{E}_{21}$ are the corresponding block matrices from the result of the exponent:

$$
\begin{bmatrix} \boldsymbol{E}_{11} & \boldsymbol{E}_{12} \\ \boldsymbol{E}_{21} & \boldsymbol{E}_{22} \end{bmatrix} \stackrel{\text{def}}{=} \exp\left( \begin{bmatrix} -\eta \boldsymbol{U}\boldsymbol{U}^T & \eta\gamma \boldsymbol{V}\boldsymbol{U}^T \\ \tau \boldsymbol{I}_{|\mathcal{D}|} & -\tau \boldsymbol{I}_{|\mathcal{D}|} \end{bmatrix} t \right) . \tag{63}
$$

Rewriting the first two matrices further yield:

$$
\begin{aligned}
&\begin{bmatrix} -\eta \boldsymbol{U}^T \boldsymbol{U} & \eta\gamma \boldsymbol{U}^T \boldsymbol{V} \\ \tau \boldsymbol{I}_{|\theta|} & -\tau \boldsymbol{I}_{|\theta|} \end{bmatrix}^{-1} \begin{bmatrix} \boldsymbol{U}^T & 0 \\ 0 & \boldsymbol{U}^T \end{bmatrix} \\
&= \begin{bmatrix} -\eta^{-1} \left( \boldsymbol{U}^T \boldsymbol{U} - \gamma \boldsymbol{U}^T \boldsymbol{V} \right)^{-1} \boldsymbol{U}^T & -\tau^{-1}\gamma \left( \boldsymbol{U}^T \boldsymbol{U} - \gamma \boldsymbol{U}^T \boldsymbol{V} \right)^{-1} \boldsymbol{U}^T \boldsymbol{V}\boldsymbol{U}^T \\ -\eta^{-1} \left( \boldsymbol{U}^T \boldsymbol{U} - \gamma \boldsymbol{U}^T \boldsymbol{V} \right)^{-1} \boldsymbol{U}^T & -\tau^{-1}\boldsymbol{U}^T + \tau^{-1}\gamma \left( \boldsymbol{U}^T \boldsymbol{U} - \gamma \boldsymbol{U}^T \boldsymbol{V} \right)^{-1} \boldsymbol{U}^T \boldsymbol{V}\boldsymbol{U}^T \end{bmatrix} \\
&= \begin{bmatrix} -\eta^{-1}\boldsymbol{U}^T \left( \boldsymbol{U}\boldsymbol{U}^T - \gamma \boldsymbol{V}\boldsymbol{U}^T \right)^{-1} & -\tau^{-1}\gamma\boldsymbol{U}^T \left( \boldsymbol{U}\boldsymbol{U}^T - \gamma \boldsymbol{V}\boldsymbol{U}^T \right)^{-1} \boldsymbol{V}\boldsymbol{U}^T \\ -\eta^{-1}\boldsymbol{U}^T \left( \boldsymbol{U}\boldsymbol{U}^T - \gamma \boldsymbol{V}\boldsymbol{U}^T \right)^{-1} & -\tau^{-1}\boldsymbol{U}^T + \tau^{-1}\gamma\boldsymbol{U}^T \left( \boldsymbol{U}\boldsymbol{U}^T - \gamma \boldsymbol{V}\boldsymbol{U}^T \right)^{-1} \boldsymbol{V}\boldsymbol{U}^T \end{bmatrix} \\
&= \begin{bmatrix} \boldsymbol{U}^T & 0 \\ 0 & \boldsymbol{U}^T \end{bmatrix} \begin{bmatrix} -\eta^{-1} \left( \boldsymbol{U}\boldsymbol{U}^T - \gamma \boldsymbol{V}\boldsymbol{U}^T \right)^{-1} & -\tau^{-1}\gamma \left( \boldsymbol{U}\boldsymbol{U}^T - \gamma \boldsymbol{V}\boldsymbol{U}^T \right)^{-1} \boldsymbol{V}\boldsymbol{U}^T \\ -\eta^{-1} \left( \boldsymbol{U}\boldsymbol{U}^T - \gamma \boldsymbol{V}\boldsymbol{U}^T \right)^{-1} & -\tau^{-1}\boldsymbol{I}_{|\mathcal{D}|} + \tau^{-1}\gamma \left( \boldsymbol{U}\boldsymbol{U}^T - \gamma \boldsymbol{V}\boldsymbol{U}^T \right)^{-1} \boldsymbol{V}\boldsymbol{U}^T \end{bmatrix} .
\end{aligned}
\tag{64}
$$

Thus, obtaining eq. (42):

$$
\begin{bmatrix} \omega_t \\ \bar{\omega}_t \end{bmatrix} = \begin{bmatrix} \boldsymbol{U}^T & 0 \\ 0 & \boldsymbol{U}^T \end{bmatrix} \begin{bmatrix} -\eta \boldsymbol{U}\boldsymbol{U}^T & \eta\gamma \boldsymbol{V}\boldsymbol{U}^T \\ \tau \boldsymbol{I}_{|\mathcal{D}|} & -\tau \boldsymbol{I}_{|\mathcal{D}|} \end{bmatrix}^{-1} \begin{bmatrix} -\eta \left( \boldsymbol{E}_{11} - \boldsymbol{I}_{|\mathcal{D}|} \right) \hat{\boldsymbol{r}} \\ -\eta \boldsymbol{E}_{21} \hat{\boldsymbol{r}} \end{bmatrix} , \tag{65}
$$

# D ENVIRONMENT EXPERIMENT DETAILS

## D.1 TOY ENVIRONMENT

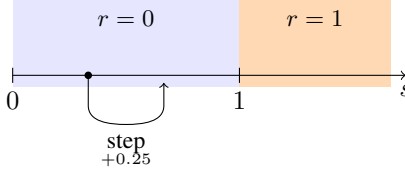

Figure 5: Illustration of the toy environment. $\mathcal{S} = \mathbb{R}_{\geq 0}$, the only action is increase the current state by 0.25. A reward of 1 is given when the state exceeds $s = 1$. An action taken from $s \geq 1$ terminates the episode.

An illustration of the environment can be seen in fig. 5. Details for the toy environment:

- State space $\mathcal{S} = \mathbb{R}_{\geq 0}$, dataset generated by getting 21 linearly spaced numbers in $[0, 1.25]$;
- Single action: $s' = s + 0.25$;
- Reward $r(s) = \begin{cases} 1 & \text{if } s \in [1, 1.25) \\ 0 & \text{otherwise}\end{cases}$;
- Episode terminates, when $s' \geq 1.25$.

The toy environment experiments were ran with the following parameters (unless specified otherwise): using the single fully connected NN case; *ReLU* activation function; $\gamma = 1$, $\eta = 0.01$, $\tau = 0.1$, 2 layers of width 1024; $2^{16}$ epochs; weights initialized with std. $\sigma_w^2 = 1$, $\sigma_b^2 = 0.1$; ensemble of 20 networks. The figures were evaluated by taking 100 linearly spaced values in the range $[0, 1.25]$.

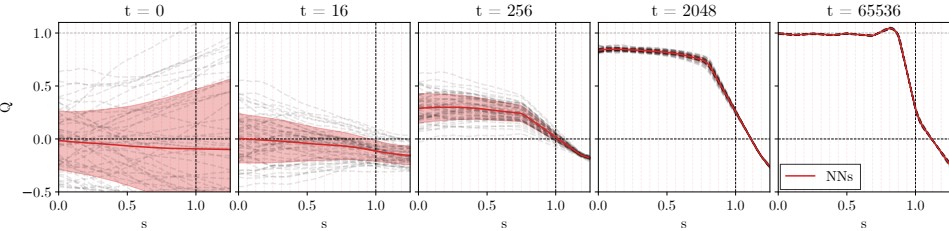

Figure 6: Individual runs of the NN from fig. 2.

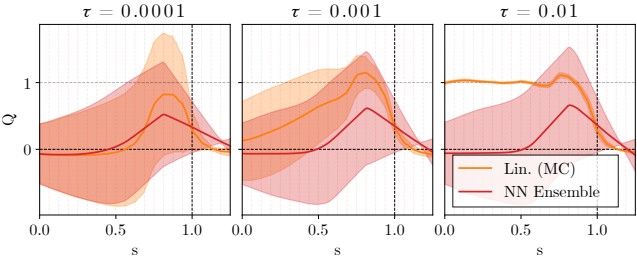

Figure 7: Comparison of different values of $\tau$ during training, at $t = 2048$. We observe that increasing the smoothing coefficient leads to faster convergence for the linearization, although the convergence of NNs seems to be unaffected. In this setting there are no theoretical guarantees that the linearized model (considering the case with a target network) is a good approximation in the infinite width. We conclude, that even though the linearized model converges to the correct Q-values, it does not accurately reflect the training dynamics of the empirical ensemble.

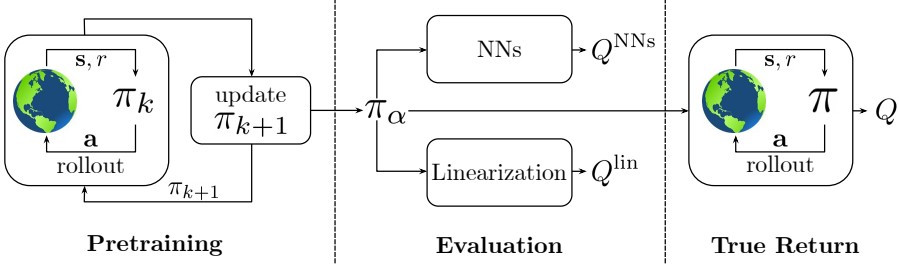

Figure 8: Schematic of the data collection and training procedures for the cartpole experiment. Figure adapted from Levine et al. (2020).

## D.2 CART POLE

In the cart pole environment (Barto et al., 1983), we train an agent to obtain a policy $\pi_\alpha$ we wish to evaluate. We then use another policy $\pi_\beta$ to generate an offline dataset of interactions with the environment (the behavior policy). In the examples in this work, $\pi_\alpha = \pi_\beta$. Our collected dataset consists of 2048 transitions.

We train a fully-connected Q-network with 2 layers of width 256, with $\gamma = 1$. The neural networks are trained in the setup with a primary Q-network only; $\eta = 0.1$, trained for $2^{16}$ epochs.

The experiment details presented in table 1 can be summarized in fig. 8. To obtain the true returns we take each element from the dataset and roll out the greedy policy based on the predicted Q-values. To obtain the negative log-likelihood, we fit the ensemble of different Q-values to a Gaussian, and use its PDF to compute the likelihoods. The presented values are the mean and std over the 2048 transitions in the dataset.

