# OpenReview forum: "Wide Neural Network Training Dynamics for Reinforcement Learning"
_ICLR.cc/2024/Conference — Submitted to ICLR 2024_

### Official Review · Reviewer_UPFL · 2023-10-27

**Soundness:** 2 fair
**Presentation:** 3 good
**Contribution:** 2 fair
**Rating:** 3
**Confidence:** 4

**Summary:**

The work presents the learning dynamics of both single value network and target network settings for residual reinforcement learning. It employs the Neural Tangent Kernel (NTK) framework to derive the learning dynamics. The authors pose this as a theoretical model to understand an ensemble of NNs trained empirically. They use the linearization (in parameters) technique for writing the dynamics of how the parameters change thereby turning the problem into a linear ODE. They also provide experimental results for a toy domain and the cartpole domain to compare the parameter dynamics and the empirically trained NN.

**Strengths:**

The paper has the following strengths:

- The authors have written it very well. I found all the technical details fairly accessible. They provide necessary background on NTK.

- The derivation of dynamics for the TD case, where you are learning using residual gradient descent, are quite informative and useful.

- They provide interesting experiments with a toy example and the undiscounted cartpole environment.

**Weaknesses:**

While I found the paper to be an interesting read I observe some crucial issues with the work which need addressing.

**Lack of related work:** While the authors have cited relevant work in DL theory, they are missing come crucial citations [1, 2, 3, 4]. The issue is that previous works have explored similar problems. For example, the setting where gradient "flows" through both $Q_{\theta}(s, a)$ and $Q_{\theta}(s', a')$ is called residual learning [3, 4]. The equation number 7 by the authors has a single sample analog in [3] (see Equation (1) in the paper). Baird [4] introduced the idea of residual algorithms to make the TD algorithm more principled. Further see related work [5]. Please let me know if I am mistaken about this connection. [1] has a different type of linearization: where they linearize the learnt function in the first layer weights for single hidden layer networks, while this is not directly applicable it has a connection with the current work. Similarly, [3] gives insights into when and if TD with NNs and homogenous activations converges albeit in discrete setting, depending on the environment.

**Not a substantial enough conclusion:**  I appreciate that the authors have derived and written down the dynamics of learning in the NTK regime. While it is a useful contribution, I am afraid that I am not able to discern a valid and strong conclusion.

I see that the authors have introduced the TD version of $\Theta_0$ (just like Lee et al. 2019) but I would be interested in how this matrix effects the convergence. I believe this is where the environment dynamics come into play. Further, I would expect a theorem similar to Theorem 2.1 by Lee et al. 2019 (latest arxiv version). Theorem 2.1 by Lee et al. 2019 provides a sense of how far the linear approximation $f^{lin}$ is from the true function $f$ i.e. order of $1/n^{-1/2}$. I would expect something similar with relation to the environment dynamics (and not only the width of the NN) for the work to be more applicable to TD learning.

Similarly, Brandfonbrener and Bruna, 2020 [2] provide some insights into when and if neural TD learning converges depending on the geometry of the space of function approximators, the structure of the underlying Markov chain, and their interaction. This brings in aspects of reinforcement learning and transition dynamics. This issue also relates to the previously stated issue of a lack of related work.

Further, the experiments seem a bit inconclusive to me, especially the experiment on cartpole domain. While I appreciate the new comparisons in Figures  3 and 4, even in a toy environment we see the parameters behaving differently (in Figure 1). The authors also remark that their analytic model does not match the empirical model in Section 5.1 but no reasoning is provided. For cartpole the result is surprising the linearized dynamics work better and is definitely a meaningful contribution. I would be curious to see results normalized by the true values? As far as I can tell, both NNs and TD-NTK do not "fit very well" with different degrees of "not fitting very well".

**Statement of assumptions:** I would be curious to know when TD-NTK does not converge? I would be surprised if it converges everywhere without assumptions (as I presume is noted in Equation 31 in the appendix). For example, [5] and [4] both say that residual RL might not always converge. Similarly, Lee et al 2019 assumed $\lambda_{min}(\Theta_0) > 0$. Further discussions regarding assumptions can be found in [1] and [2]. These assumptions are better stated clearly.

I do not intend to discourage the line of work the authors are pursuing. It is very important for the research community that we have a concrete theory of Deep RL that relates to practice. The training dynamics derivation is very useful and much appreciated but the work would fair better with a stronger conclusion and connections to previous work.

****

**References**

[1] Neural Temporal-Difference and Q-Learning Provably Converge to Global Optima, Qi Cai, Zhuoran Yang, Jason D. Lee, Zhaoran Wang, 2019

[2] Geometric Insights into the Convergence of Nonlinear TD Learning, David Brandfonbrener, Joan Bruna, ICLR 2020

[3] Deep Residual Reinforcement Learning, Shangtong Zhang and Wendelin Bohmer and Shimon Whiteson, 2019

[4] Residual algorithms: Reinforcement learning with function approximation, ICML 1995

[5] Gradient temporal-difference learning algorithms, Richard S. Sutton and Hamid Reza Maei, 2011

**Questions:**

**Questions:**
While most questions are above I present a few more here:
1. What is the state space in the cartpole environment?
2. How was the behavior policy learnt in cartpole environment?
3. What is the idea behind deriving the expression for $Q^{lin}(\mathcal X^*)$? Is this related to generalization (I guess similar to Lee et al. 2019)?
4. If you consider the central graph in Figure 1, it shows that the difference in parameter dynamics is substantial.
5. I would expect that the comparison should be if the $Q$ functions predict the right action (as opposed to the accuracy), across empirical and analytic models, as it is well known that a "wrong" NN Q function can lead to a good policy as seen in Figure 3 of the DDPG paper (arxiv version). This would be interesting in addition to Figure 2. Same applies for table 1.


**Minor issues:**

1. Section 2.1 we will complement -> we complement
2. Ideally, I wouldn't reference Equation 6 from Section 4.1 under Equation 2 in Section 3.
3. Section 3: forgot an opening bracket for "such as a sigmoid or ..."
4. $\mathcal X$ might be better defined as a matrix than a "stack"?
5. $\theta$ -> $\theta_t$ in the caption below the expression for $Q^{lin}_{\theta_t}$. Also, missing a comma after the expression.
6. $w$ is overloaded in section 2: $w_t in Equation 5 and $w_{i,j}$ below Equation 3.
7. Missing full stop at the end of equation 6.

---

> ### Author Response · Authors · 2023-11-22
>
> We are very thankful for the thorough feedback and constructive, actionable feedback provided. We appreciate the reviewer’s recognition of the value in finding solutions to the temporal difference dynamics and interest in its related empirical investigation. We thank the reviewer for providing relevant references to support this work and would like to address some of the points raised.
>
> We have not included a thorough derivation of the approximation $f^\text{lin}$ distance to $f$ similar to theorem 2.1 by Lee et al. (2019); as we describe in the text (section 4.1, after eq. 6) why the reasoning is analogous if the equivalent assumptions from Lee et al. are made in the temporal difference setup, and the same results, such as errors of the order $1/n^{-1/2}$ with hidden layer with $n$, still holds.
>
> We do agree with the reviewer that the environment dynamics manifest themselves in $\Theta_0$ which, as we point out in eq. 11, would affect what the critical learning rate for convergence is. As far as training dynamics are involved, we don’t think the environment dynamics (at least in the deterministic dynamics case) introduce additional caveats towards the conditions necessary for our approximations to hold.
>
> Answers to Questions:
> The cartpole environment has a 4-dimensional state space (position of the cart, velocity of the cart, angle of the pole, angular velocity of the pole). We use 2048 transitions $(s, a, r, s’)$ to train the networks.
> The behavior policy was a $\max_{a’} Q(s’, a’)$ from a Q-function learned by a single hidden-layer DQN agent trained online.
> Finding an expression at arbitrary state-actions $\mathcal{X}^*$ is indeed to investigate the generalization of the predictions of the model/ensemble. The high-level idea to derive this is that by the linearisation of $Q(\mathcal{X^*}) = Q_0(\mathcal{X}^*) + \nabla_\theta Q_0(\mathcal{X}^*)\omega_t$, one can substitute the expression for $\omega_t$ to obtain the expression for Q-values at arbitrary state-actions $\mathcal{X}^*$ throughout and at the end of training.
> That is true, some degree of discrepancy is expected to accumulate throughout training.
> The environment represented in figure 2 is the toy environment, having only one action, although we can include this for the cartpole environment
> Minor issues: we have incorporated the suggested changes into the paper.

---

### Official Review · Reviewer_5Tmm · 2023-10-29

**Soundness:** 1 poor
**Presentation:** 2 fair
**Contribution:** 1 poor
**Rating:** 1
**Confidence:** 4

**Summary:**

The paper attempts to apply the infinite-width neural network learning via the neural tangent kernels (NTK) approach recently introduced in the supervised learning setting to an (offline) reinforcement learning (RL) setting.
The paper first derives some closed-form expressions for the training dynamics of the Q-network parameters using the NTK approach. Then, experimental evaluation using a custom toy environment and the cart-pole environment is provided to demonstrate the effects of uncertainty quantification of Q-network ensembling.

**Strengths:**

1. The paper attempts to study an ambitious and non-trivial problem of applying the infinite-width neural networks approach known in supervised learning to the RL setting.
2. Some promising results are demonstrated in the reported preliminary experimental evaluation, which can be the basis of a future more extensive study.

**Weaknesses:**

Although the studied research direction is interesting and promising, the paper has some significant flaws, and the contribution of the paper is weak in its present form. The study requires a significant extension to be further considered for publication.

1. The theoretical contribution of the paper is merely the calculation of NTK dynamics for Q-learning by supervised learning over fixed datasets of rollouts adapted from existing work [ Lee et al. (2019) ]. This part does not bring any new contributions or insights.

2. The title, abstract, and introduction of the paper need to be more accurate.
Setting studied in the paper are not those of online or offline reinforcement learning. The studied setup can be characterized as a Q-learning approach via supervised learning over a fixed dataset. Namely, a dataset of dumped policy-environment interactions is used to train the parameters of a Q-network. It differs from the standard online and offline actor-critic RL setups in which the actor (policy) and critic (Q-network) is approximated simultaneously.
Namely, the work ignores the issue of approximating the policy action on the next-state; this kind of analysis detached from policy properties is a pure Q-learning.
As stated, the Bellman bootstrap equation reads $Q^{\pi}(s, a) = r + \gamma \sum_{s'} P(s'|s,a) \pi(a'|s') Q^{\pi}(s', a')$, but it does not lead to the supervised learning loss function (6). $\mathcal{X}'$ is unavailable in the RL setup. The dataset MDP tuples  $ (s_i, a_i, r_i, s_i')$ does not contain the next actions; they should be obtained from the policy instead, as the Q is an estimated value of the trajectories obtained _by following the policy $\pi$_. It is a separate issue of how to regularize and avoid issues with distribution shift at this stage addressed by modern Offline RL methods, including TD3-BC [1] and FDCR [2].

[1] Kostrikov I., "Offline Reinforcement Learning with Fisher Divergence Critic Regularization", ICML 2021
[2] S. Fujimoto and S. Gu, "A Minimalist Approach to Offline Reinforcement Learning", NeurIPS 2021

3. Experimental evaluation could be stronger. Only simple environments (a toy one and a cartpole are used) are particular, and it is unclear how the obtained conclusions will transfer to more complex environments widely used as benchmarks (MuJoCo).

Minor remarks:
* p.5 below (11) "maximum and minimum values of Θ." -> eigenvalues,
* p.5 below (11) "This is expected because in this regime learning the value function is essentially a
supervised learning task of learning the immediate rewards, and all terms with policy dependence
(those involving X ′ ) drop out as would be expected in a supervised learning task." in supervised learning task that is true, but unfortunately not in an RL task :)
* p. 6 (5.1) "the value will be $1$ (taking $\gamma = 1$) for all $s < 1$ and $0$ otherwise." I think the condition is switched here (value $0$ for $s<1$).
* p. 8 (5.2) "the true returns obtained by rolling out the evaluated policy in the environment to evaluate our method." the issue of policy-generating data requires more attention in the experimental section.
* derivations in C.1 and C.2 are not needed, these are solutions of simple non-homogenous differential equations,

**Questions:**

See above. At present I think that the current paper needs to be framed as a supervised Q-learning  to convey a proper message. Offline RL setting would require to include the policy in the analysis. The experimental evaluation need to be extended beyond toy-problems to show that the conclusions are true for larger scale benchmark problems and NTK is a proper tool for studying the statistical properties of Q-ensembles.

---

> ### Author Response · Authors · 2023-11-22
>
> We appreciate the time and effort the reviewer has committed to providing constructive feedback on our manuscript. We are grateful for the recognition of the ambitious nature of the problem, and that the findings were perceived as promising. We would like to address some of the concerns pointed out in the review.
>
> **Weakness 1:**
>
> We want to first maintain that our work is distinct from supervised learning. We believe it is inaccurate to frame our approach as “supervised Q-learning” and disagree with the reviewer in our theoretical analysis not providing novel contributions. Supervised learning assumes ground-truth labels available for the quantity that is being learned, and we stress is not the case in our setup: we do not assume no return samples or extended traces are available (only states, actions, next-states and immediate rewards for single transitions are observed). The novelty in our contribution is exactly to describe how to adapt the methods that analyze training dynamics from the neural tangent kernel perspective in supervised learning to the temporal difference policy evaluation setting with only $(s,a,r,s’)$ tuples available. Crucially, the policy being evaluated does not need to coincide with the policy that generated the data.
>
> **Weakness 2:**
>
> Again, we disagree that the setting considered is supervised learning as no ground-truth return labels are available for training. We agree with the reviewer that we are not directly modeling the full “reinforcement learning” task, and this is highlighted in the abstract where we specifically describe our contributions as being relevant to “temporal difference policy evaluation methods” and associated uncertainty quantification. Nonetheless, forms of temporal difference off-policy evaluation are key elements present in many RL algorithms, and we believe better understanding temporal difference policy evaluation with neural networks is an important milestone towards better understanding deep RL (particularly in the offline setting with a static dataset).
>
> We would also like to point out that, as specified in section 3, the actions in the state-action pairs $\mathcal{X’}$ are those computed from the policy *that we wish to evaluate* at the next-states, which is assumed given in the policy evaluation setting and would correspond to the action taken by the actor in the RL setting (in the RL setting, this quantity is also derived from the actor). In fact, we entirely agree with the reviewer’s comment that “Q is an estimated value of the trajectories obtained *by following the policy* $\pi$”. As the action in $\mathcal{X’}$ is taken from the policy we are evaluating, and is not related to the next actions taken in the trace that generates the data, these are exactly the Q-values learned by training with the loss in eq. 6.
>
> **Weakness 3:**
>
> While we appreciate that this is not confirmed with empirical experiments in the current version of our work, our focus was on exposing the theory, and much like previous work investigating neural tangent kernel training dynamics we do not expect size of state or action space size to have a significant effect on the quality of the training dynamics approximations
>
> **Minor remarks:**
>
> We would like to note, that for the setting where $\gamma = 0$ (minor remark #2), the equation simplifies to the supervised learning case. The other case, when $\gamma \neq 0$ it is no longer a supervised learning task, and we wanted to highlight this difference in the text. For minor remark #3 we maintain that the true value function will be 1 when $s$ is less than 1 ($V(s) = 1$ when $s < 1$). The agent always moves right, and crossing the line at $s=1$ grants a reward of 1, so everything on the left of the line ($s<1$) has value 1 (when $\gamma=1$) and everything to the right ($s>1$) leads to terminal states with no further rewards, and the value here must be 0.

---

### Official Review · Reviewer_bQbn · 2023-11-01

**Soundness:** 3 good
**Presentation:** 2 fair
**Contribution:** 2 fair
**Rating:** 3
**Confidence:** 3

**Summary:**

This paper studies the training dynamics of linearized neural network (NTK approximation) temporal difference learning. The authors give solutions to the differential equations for full-batch gradient flow in two settings: learning with a single Q-network, and learning with a separate target network which is an exponential moving average of the primary Q-network. They also give closed-form solutions for the uncertainty from an ensemble of randomly-initialized neural networks under the NTK approximation in these settings. The analytical solutions are compared to actual neural networks trained on two toy domains include CartPole.

**Strengths:**

The paper is generally clear and easy to follow. The derivation of the training dynamics for TD learning seems like it could be useful for understanding deep RL theoretically. I am not very familiar with the area but from some searching on Google Scholar the analysis appears to be novel. The results in toy domains suggest that the theoretical results are predictive of real neural network training.

**Weaknesses:**

While in general the paper presents useful findings, I think it may lack enough overall contribution for publication at ICLR. It's not clear what the key takeaways are from the paper beyond the differential equation solutions, which are closely based on prior work. Some specific reasons that I feel there is not enough contribution:
 * The theoretical results are presented without much explanation or intuition. It's not clear if there are any insights that can be gained from the solutions to the differential equations.
 * The empirical results are only for the simplest of RL environments (1 and 4 dimensional state spaces; 0 and 1 dimensional action spaces). Since deep RL is primarily useful for high-dimensional environments with complex value functions, it would be more convincing if the authors evaluated on these types of environments as well.
 * The analysis is restricted to finding the Q-function of a fixed policy, while reinforcement learning algorithms generally use a TD loss like $\left(Q(s, a) - (R(s, a) + \\max_{a'} Q(s', a'))\right)^2$.

Also, there has been some previous analysis of RL with NTK approximation by Yang et al. [1], who go beyond training dynamics for a fixed-policy and give regret bounds for an optimistic variant of Q-learning with the NTK. It would be good to include a comparison to their work.

[1] Yang et al. On Function Approximation in Reinforcement Learning: Optimism in the Face of Large State Spaces. NeurIPS 2020.

**Questions:**

Related to the weaknesses listed above:
 * What insights can be drawn from the analytical solutions for NTK TD learning gradient flow?
 * Are the analytical solutions presented in this paper also good approximations for TD learning in higher-dimensional state/action spaces?
 * Can the analysis be extended to using a fitted Q-iteration loss where $Q(s', a')$ is replaced with $\\max_{a'} Q(s', a')$?

---

> ### Author Response · Authors · 2023-11-22
>
> We are grateful to the reviewer for their time and effort in providing constructive feedback to our work. We also appreciate the author acknowledging the novelty in our analysis and the potential it brings for insight into deep RL theory.
>
> **Question 1 (What insights can be drawn from the analytical solutions for neural tangent kernel TD learning gradient flow?):**
>
> The regime (i.e., when learning rate $\eta < \eta_\text{critical}$) suggests that many NNs used in practice may be operating in the setting we are investigating. Also, the derived kernel $\Theta$ captures the training dynamics of these settings, and can be studied in the future for insight into stability and generalization. We think that analyzing the behavior of infinitely-wide NNs offers a foundation for developing a systematic comprehension of finite-width NNs.
>
> **Question 2 (Are the analytical solutions presented in this paper also good approximations for TD learning in higher-dimensional state/action spaces?):**
>
> The derived solutions in theory are general, and also apply to higher-dimensional state/action spaces. While we appreciate that this is not confirmed with empirical experiments in the current version of our work, our focus was on exposing the theory, and much like previous work investigating neural tangent kernel training dynamics we do not expect size of state or action space size to have a significant effect on the quality of the training dynamics approximations
>
> **Question 3 (Can the analysis be extended to using a fitted Q-iteration loss where $Q(s’, a’)$ is replaced with $\max_{a’} Q(s’, a’)$?):**
>
> We appreciate that this is a very relevant extension to our work which we are actively investigating, and that it may introduce additional insight into the instabilities present in offline RL related overestimation of Q-values.

---

> > ### Comment · Reviewer_bQbn · 2023-12-04
> > **Response to authors**
> >
> > I thank the authors for their response to my comments. I have decided to maintain my score since I think that while this paper shows promise it is not yet ready for publication. I agree with the other reviewers that the setting the authors consider quite far from RL in practice, limiting its applicability. The experiments are also somewhat limited. Finally, there is a lack of significant discussion of the implications of the theoretical results. The paper could be improved for future submissions by addressed some of these weaknesses.

---

### Official Review · Reviewer_GToL · 2023-11-01

**Soundness:** 2 fair
**Presentation:** 2 fair
**Contribution:** 2 fair
**Rating:** 3
**Confidence:** 4

**Summary:**

This work compares the training dynamics of TD-learning with neural networks with it's NTK parameterisation. It does this for two settings, a full-gradient one in which it minimizes the mean squared TD error and also for the more common, semi-gradient with target network scenario.

It finds evidence that the lazy regime dynamics only matches in the full-gradient setup. In addition, for the same setup, it finds evidence that the linearized parameterization can provide more accurate uncertainty estimates than SGD ensemble training.

The study is concerns TD policy evaluation with a fixed policy and the empirical evaluation is performed on cart-pole and a single-action, 21-states MDP.

**Strengths:**

I find the proposal of using tools for understanding of neural network training dynamics from the supervised learning setup to the TD-learning policy evaluation scenario interesting and potentially useful in identifying the peculiarities of optimisation in RL.

All of the empirical observations on the adequacy of the linearised model for describing the training dynamics and the quality of uncertainty estimation of ensembles can be useful for developing full-gradient mean TD-error algorithms (naive residual-gradient algorithms), although it remains debatable whether this is a proper objective in RL.

I found the writing and notation to be generally clear and consistent.

**Weaknesses:**

The paper mainly focuses on the training dynamics of full-gradient TD-learning, which, to my understanding, is similar to minimising the average TD-error. This severely limits the potential impact of the work since full-gradient algorithms are not used in practice and, more importantly, it's not entirely clear if the solutions to the objectives in full-gradient algos are actually desirable (see eg. chapters 11.4 - 11.5 in Sutton). I believe this aspect should be better highlighted and contextualized in the paper.

Authors also show that their NTK derivation of TD-learning with semi-gradient descent and target network (the more relevant setup) does not follow the same dynamics as under SGD which is an interesting finding even if negative. However the relevant figures are pushed to the appendix and there is not discussion as to why might this be the case. Is it because NTK simply is not a good approximation for algorithms which do not follow a proper gradient field? Is it because of the added complexity of the slowly moving target network?

The last question also points to some lack of clarity in the paper as it bundles together target networks with semi-gradient methods. However having a different target function is a separate design decision from choosing not to use the gradient of the target network. This also leads to a possible experiment: is the linearisation better at approximating training dynamics in the semi-gradient setup without a separate target network?

Finally, it would be interesting to mention and develop how your empirical findings impact some of the previous works that use the linearisation of neural (gradient/semi-gradient) TD-learning to come-up with convergence proofs. Examples include:
- https://proceedings.neurips.cc/paper/2020/file/75ebb02f92fc30a8040bbd625af999f1-Paper.pdf
- https://arxiv.org/pdf/1912.04511.pdf

**Questions:**

Questions and actionable insights in the weakness section.

---

> ### Author Response · Authors · 2023-11-22
>
> We are thankful to the reviewer for the thorough and constructive feedback of our work, and appreciate the interest the reviewer acknowledges in our analysis as a potential tool for better understanding the challenges in RL optimisation. We would like to address here some of the points raised in the review.
>
> We would like to note that the toy environment has a continuous state space, and we chose to train the agent from 21 starting positions (instead of a 21-states MDP).
>
> **Concern 1 (are the solutions to the objectives in full-gradient algos actually desirable?):**
>
> While we agree that full-gradient algorithms are not yet often used in practice, it is a common simplifying assumption in analyzing training dynamics (for example in the neural tangent kernel literature) to retain a degree of theoretical tractability while still maintaining the core ingredients of the optimization process involved. While we appreciate the example highlighted by the reviewer as to why the stated gradient objective may not always be desirable, these are valid for some stochastic MDPs, and in many other cases, including all environments with deterministic transitions, the training will converge to a consistent value function. We have amended the manuscript to discuss these two points, and therefore explain why our approach is still sound for many environments of interest.
>
> **Concern 2 (no discussion on why TD semi-gradient with target network does not follow the same dynamics as SGD):**
>
> In section 4.1, we justify why the linearisation holds for the single network case. The same reasoning does not directly hold for the target network analysis in section 4.2, and as such there is no direct theoretical guarantee that the linearisation will also hold here. We included the analogous analysis and empirical results anyway due to the relevance of the setup to practical algorithms, which we are glad the reviewer agrees are interesting negative results. We have amended the manuscript to clarify and put in better context how the analysis for the target network case is not guaranteed to approximate the true learning dynamics in the same way the single network case does.
>
> **Concern 3 (target networks and semi-gradient methods bundled together):**
>
> We considered including an analysis for the setting suggested by the reviewer that employs semi-gradient methods without a target network. However, we decided not to because being a semi-gradient it does not satisfy the conditions for neural tangent kernel linearistation to hold (unlike the case analyzed in section 4.1) and is also not common in practice.

---

### Meta-Review · Area_Chair_v9xb · 2023-12-09

**Metareview:**

This paper dervies differential equations for understanding the behavior of TD-learning for policy evaluation with wide neural networks. And then utilize this analysis for understanding behaviors of ensembles and uncertainty in TD-learning. Overall, while I find the goal and motivation to be quite important to study in deep RL, this paper falls short of making a clear contribution. As several reviewers indicate, several prior works derive training dynamics for deep RL algorithms, including with NTK (another example: https://openreview.net/forum?id=shbAgEsk3qM), and these were not discussed. Likewise, the Lyle et al. 2019 paper is not discussed well, though from my understanding it also derives some of these equations. It is also unclear what the implications of the results should be and the empirical results are not concrete in my opinion to justify the claims. Overall, I would encourage the authors to consider addressing the reviewers concerns, and re-submit an expanded / detailed version of the paper, which appropraitely describes the contributions and implications to the next conference.

**Justification For Why Not Higher Score:**

Insufficient contribution, discussion of prior work, and implications.

**Justification For Why Not Lower Score:**

N/A

---

### Decision · Program_Chairs · 2024-01-16

Reject